# Housing temperature influences exercise training adaptations in mice

Steffen H. Raun [1], Carlos Henriquez-Olguín [1], Iuliia Karavaeva[2], Mona Ali[1], Lisbeth L. V. Møller [1], Witold Kot[3], Josué L. Castro-Mejía [4], Dennis Sandris Nielsen [4], Zachary Gerhart-Hines[2], Erik A. Richter [1✉] & Lykke Sylow [1✉]

Exercise training is a powerful means to combat metabolic diseases. Mice are extensively used to investigate the benefits of exercise, but mild cold stress induced by ambient housing temperatures may confound translation to humans. Thermoneutral housing is a strategy to make mice more metabolically similar to humans but its effects on exercise adaptations are unknown. Here we show that thermoneutral housing blunts exercise-induced improvements in insulin action in muscle and adipose tissue and reduces the effects of training on energy expenditure, body composition, and muscle and adipose tissue protein expressions. Thus, many reported effects of exercise training in mice are likely secondary to metabolic stress of ambient housing temperature, making it challenging to translate to humans. We conclude that adaptations to exercise training in mice critically depend upon housing temperature. Our findings underscore housing temperature as a critical parameter in the design and interpretation of murine exercise training studies.

[1] Section of Molecular Physiology, Department of Nutrition, Exercise and Sports, Faculty of Science, University of Copenhagen, Copenhagen 2100, Denmark. [2] Novo Nordisk Foundation Center for Basic Metabolic Research, University of Copenhagen, Copenhagen 2200, Denmark. [3] Department of Plant and Environmental Sciences, Faculty of Science, University of Copenhagen, Frederiksberg 1871, Denmark. [4] Department of Food Science, Faculty of Science, University of Copenhagen, Frederiksberg 1958, Denmark. ✉email: Erichter@nexs.ku.dk; Lshansen@nexs.ku.dk

Physical inactivity is a leading cause of morbidity and premature mortality worldwide[1,2] and is associated with insulin resistance, obesity, and loss of muscle mass[3]. Regular exercise training (ET) is one of the most powerful means to combat diseases by eliciting health benefits on nearly all organ systems of the body[4–6]. Thus, much effort has been targeted towards understanding the underlying molecular mechanisms responsible for the adaptive responses to ET. In such studies, mice are extensively used as an experimental tool. However, failure to recognize that the laboratory mouse is housed under mild cold stress at ambient temperature[7,8] may confound data interpretation and translatability to humans, who primarily live at thermoneutrality[9–11]. In light of this, we hypothesize the optimal conditions for performing exercise studies in murine models must be reconsidered. Specifically, maintaining mice in thermoneutral conditions could avoid chronic cold stress and its potential effects on adaptations to ET.

Mice prefer housing temperatures at 30 °C compared to 20 and 25 °C[12]. Indeed, mice housed at ambient temperatures for longer periods of time experience adverse effects on overall metabolic health[13]. They display elevated basal metabolic rate[14], have twofold increased heart rate compared to mice housed at thermoneutrality[15], show non-shivering thermogenesis due to increased sympathetic drive and activation of brown adipose tissue[8], and exist right at the cusp of immune suppression[16]. Yet the influence of housing temperature on these critical parameters is often underappreciated. Nevertheless, markedly different results have been obtained when testing the same procedures in mice housed at various temperatures, notably including mitochondrial uncoupling[17], whole-body glucose tolerance[18,19] (although recently contradicted by ref. [20]), inflammation[21], immune responses[22], age-related bone loss[23], atherosclerosis[24], and cancer[25,26]. Optimal murine housing conditions to better mimic the metabolic rate of humans have been put forth in recent years with temperatures from 27 °C[27] to 30 °C[28,29] being reported as an optimal temperature range. However, our understanding of how thermoneutrality affects the adaptations to ET is limited.

Here we undertake a detailed comparison of an extensively used mouse ET model in ambient temperature (22 °C) or thermoneutral temperature (30 °C). We show that housing temperature markedly influences the molecular and organismal responses to voluntary ET and that many reported effects of training in mice could be secondary to the metabolic stress of 22 °C housing. Consequently, our work holds broad implications for the interpretation of results, future study design, and the translatability of this model to the human condition.

## Results

### Ambient housing temperature increases energy expenditure and metabolic fluctuations in mice at rest and during exercise.

To elucidate the effect of housing temperature on exercise adaptations in mice, we housed mice at 22 or 30 °C with ET or without (untrained; UT) free access to a running wheel for 6 weeks (excl. a 7–10 days temperature acclimatization period) (Fig. 1a). This is one of the most commonly used training models for rodents often denoted 'voluntary wheel running'. Body weight gain was similar between all groups during the 6 weeks intervention (Fig. 1b, Supplementary Fig. 1a). To determine the impact of housing temperature on whole-body adaptations to ET, we conducted a series of experiments in indirect calorimetry chambers.

We first sought to determine the effects of temperature on whole-body metabolism in mice already trained for 5 weeks (at 22 or 30 °C) by placing them in the metabolic chambers with or without access to a running wheel. Voluntary ET increased

nightly $VO_2$ compared to UT mice in both temperatures, but the effect of nighttime in ET was 60% higher at 22 °C compared to 30 °C housed mice (Fig. 1c, d). As expected, RER showed diurnal rhythm at 22 °C, where RER during the day was reduced in ET compared to UT mice (Fig. 1e). When housed at 30 °C, RER was similar between day and night in UT mice, in contrast with previous reports[20,28], with no effect of ET on resting RER (Fig. 1e).

We next investigated to what extent and how rapidly the metabolism of mice changes when increasing housing temperature from 22 to 30 °C. Gradually raising the temperature over ~3 h caused a rapid drop in oxygen uptake ($VO_2$; Supplementary Fig. 1b) and respiratory exchange ratio (RER; Supplementary Fig. 1c) within 6 h of temperature change. The change in temperature led to a decrease (−45%) in food intake (Supplementary Fig. 1d) without any acute changes in habitual activity (Supplementary Fig. 1e). These findings illustrate that housing temperature robustly and rapidly alters mouse metabolism, aligning with previous reports[9,10,28,30], and underscore the metabolic challenges that are imposed on mice housed at ambient temperatures. Chronically housing mice at thermoneutrality increased habitual activity during the night by +90% in UT mice (Fig. 1f) underlining the importance of adequate acclimatization time during temperature changes, as acute temperature increase did not change habitual activity (Supplementary Fig. 1e).

Throughout the training intervention, mice housed at 30 °C ran 60% of the distance completed by mice housed at 22 °C (Fig. 1g). Using the metabolic cages, the nightly running volume tended to remain lower ($p = 0.064$, student's $t$-test, −35%, Fig. 1h) in mice housed at 30 °C compared to ambient temperature while running volume during daytime was doubled at 30 °C (Fig. 1i). Still, the distance during daytime only accounted for a small fraction of the total running volume in both temperatures. This was accompanied by a tendency towards a lower maximal ($p = 0.054$, student's $t$-test, Fig. 1j) and decreased average (Fig. 1k) running speed. To test if reduced running volume could be due to heat stress during exercise, we measured core temperature during the day (when the mice are resting/inactive) and in the early dark period (when the mice are running the most). UT mice displayed 0.6 °C lower core temperature during the inactive period when housed at 22 °C compared with 30 °C, highlighting the mild cold stress inflicted by 22 °C housing (Fig. 1l). Core temperature increased to the same absolute values in all groups during the dark period (Fig. 1l). Thus, reduced running of mice housed at 30 °C is likely not due to overheating.

Overall, mice housed at 22 °C displayed increased exercise-induced and diurnal metabolic fluctuations as well as augmented running volume compared with mice at 30 °C.

### Exercise-induced changes in body composition and metabolic improvements are enhanced at ambient temperature.

We next asked whether housing temperature would affect the metabolic benefits to voluntary ET. Having established remarkable differences in ET volume with this model, we included a group of mice with restricted voluntary running housed at 22 °C (Fig. 2a). This mimics the training volume of thermoneutrally housed mice (the running volume-paired ET groups will be denoted, "paired 22 °C ET" onwards, Fig. 2b) and enabled us to determine if any differences observed were due to the reduced running volume, or the housing temperature per se.

Despite the differences in running distance, ET elicited the same improvements (+35%) in maximal running speed (Fig. 2c) in all groups. As expected, UT mice did not improve performance during the intervention period (Supplementary Fig. 2a). Food

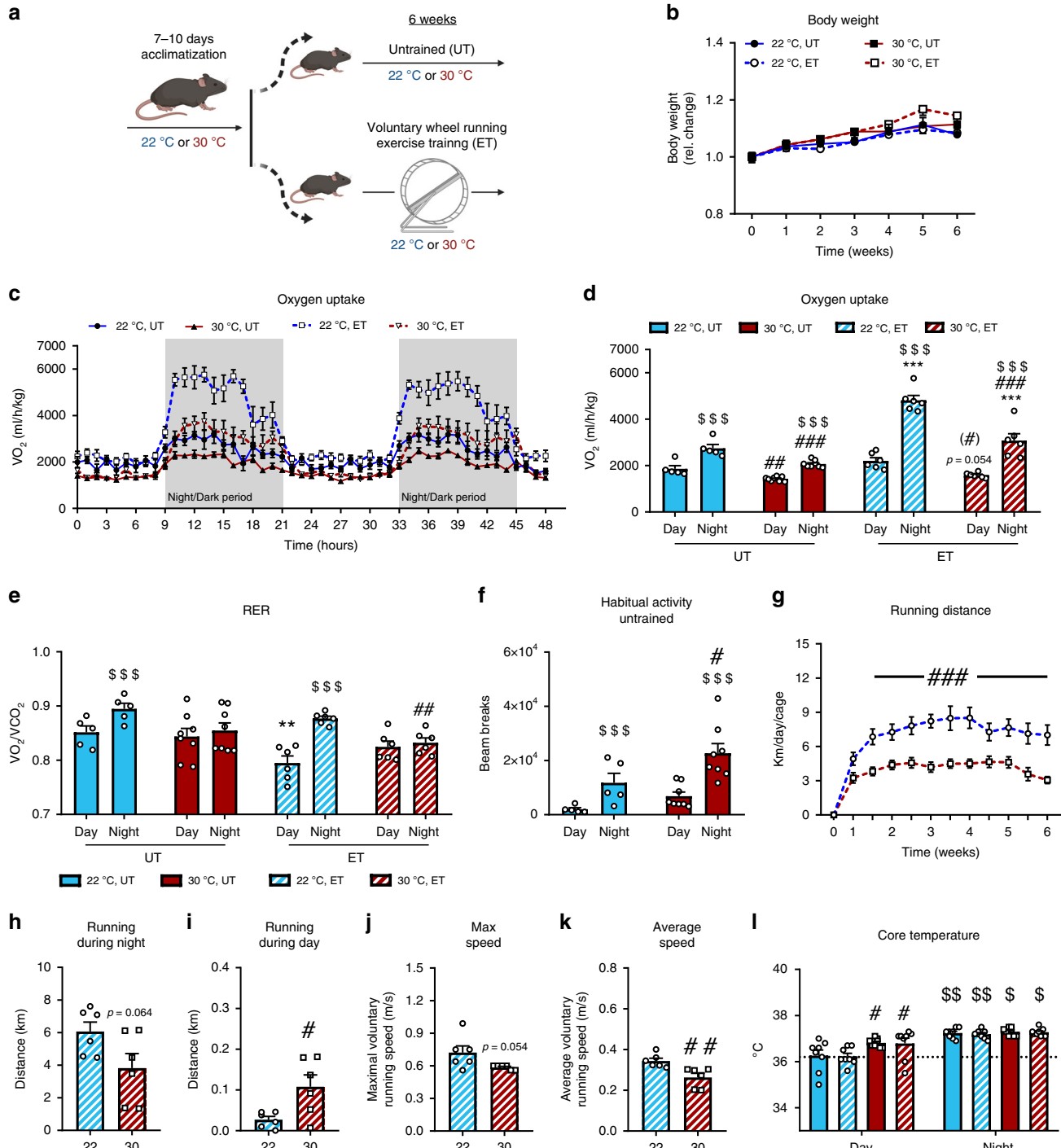

**Fig. 1 Thermoneutral housing lowers energy expenditure and metabolic fluctuations in exercising mice. a** Graphic illustration of the experimental exercise training model. Mice were acclimatized to housing temperature before completing a 6 weeks voluntary wheel running exercise training (ET) intervention. (UT = untrained). **b** The effect of housing temperature and ET at 22 °C and 30 °C on body weight. Statistical testing; Two-way ANOVA (repeated measures). 22 °C UT; $n = 8$, 22 °C ET; $n = 10$, 30 °C UT; $n = 9$, 30 °C ET; $n = 10$. **c–e** VO$_2$ and RER in UT and ET mice at 22 °C and 30 °C housing. Statistical testing; Two-way ANOVA (repeated measures). 22 °C UT; $n = 5$, 22 °C ET; $n = 6$, 30 °C UT; $n = 8$, 30 °C ET; $n = 6$. Effect of time within group; $^{\$\$\$}p < 0.001$. Effect of temperature within time of day; $^{\#}p < 0.05$, $^{\#\#}p < 0.01$, $^{\#\#\#}p < 0.001$. Effect of ET within temperature; $^{**}p < 0.01$, $^{***}p < 0.001$. **f** Habitual activity (2 consecutive days) in UT mice after 6 weeks temperature acclimatization. Statistical testing; Two-way ANOVA (repeated measures). 22 °C UT; $n = 5$, 30 °C UT; $n = 8$. Effect of time within group; $^{\$\$\$}p < 0.001$. Effect of temperature within time of day; $^{\#}p < 0.05$. **g** Running distance per day at 22 °C and 30 °C, respectively. Statistical testing; Two-way ANOVA (repeated measures). $n = 10$. Effect of temperature; $^{\#\#\#}p < 0.001$. **h–k** Effect of temperature on wheel running distance, maximum and average speed. Statistical testing; student's $t$-test. 22 °C ET; $n = 6$, 30 °C ET; $n = 6$. Effect of temperature; $^{\#}p < 0.05$, $^{\#\#}p < 0.01$. **l** Core temperature was measured at day (light period) and night (dark period) time via rectal thermometer. Statistical testing; Two-way ANOVA (repeated measures). $n = 8$ Effect of time, day vs. night; $^{\$}p < 0.05$, $^{\$\$}p < 0.01$. Effect of temperature within day; $^{\#}p < 0.05$. Data are presented as mean ± SEM incl. individual values where applicable. The "n = x" defines the number of biologically independent animals used for the analyses.

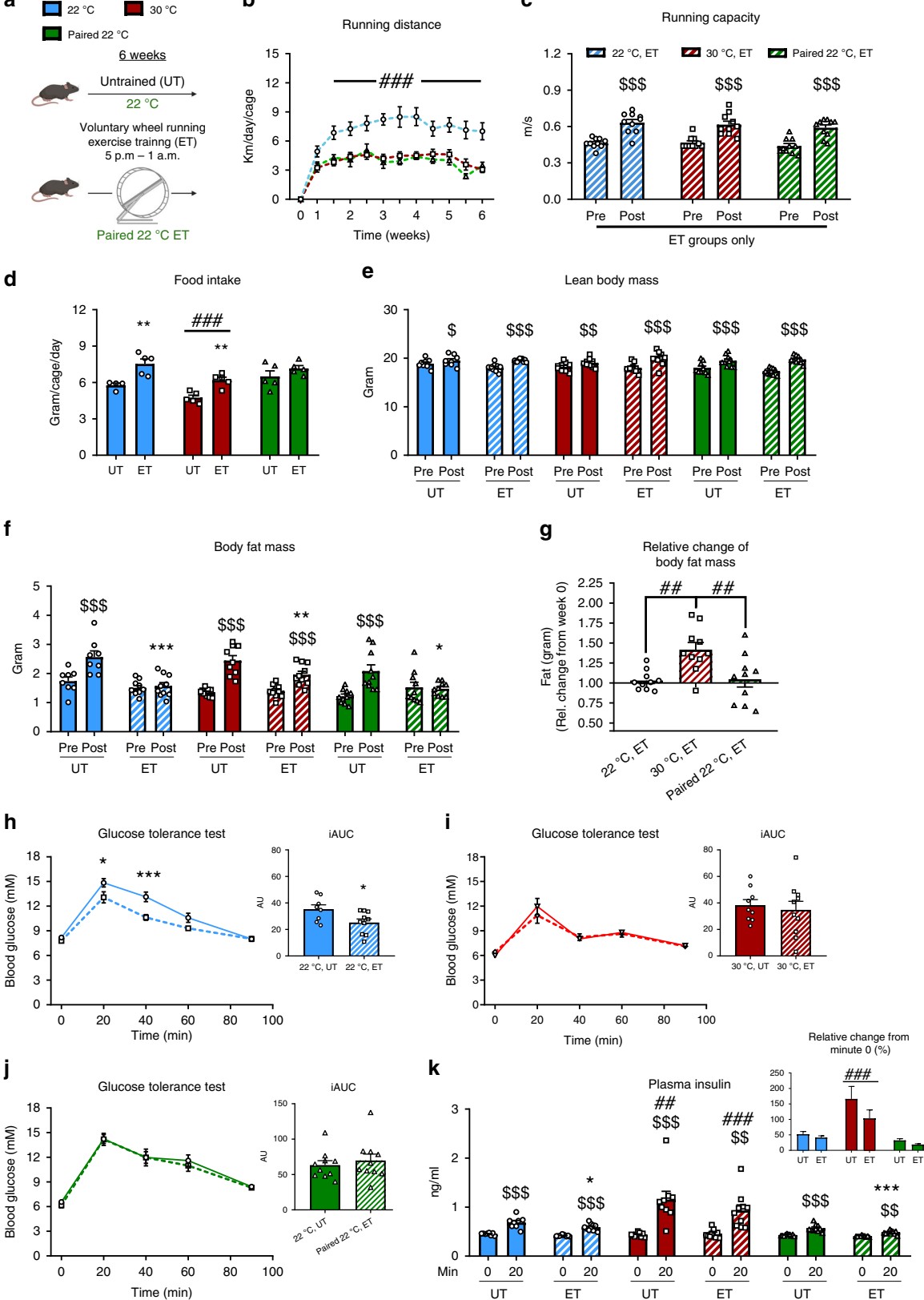

**Fig. 2 Exercise-induced changes in body composition and metabolic improvements are reduced at thermoneutrality. a** Graphical illustration of the paired 22 °C voluntary wheel running exercise training (ET) intervention. **b** Running distance per day in 22 °C, 30 °C, and paired 22 °C, respectively. Statistical testing; Two-way ANOVA (repeated measures). 22 °C ET; $n = 10$, 30 °C ET; $n = 10$, paired 22 °C ET; $n = 5$. Effect of temperature; [###]$p < 0.001$. **c** Exercise capacity before (Pre) and after (Post) the training intervention in ET groups only. Control mice (UT) are shown in Supplementary Fig. 2a. Statistical testing; Two-way ANOVA (repeated measures). $n = 10$. Effect of time within group; [$$$]$p < 0.001$. Effect of ET within temperature; [***]$p < 0.001$. **d** The effect of housing temperature and ET at 22 °C and 30 °C on food intake. Values are the average over 3 days of 3 different weeks from 4–5 cages. Statistical testing; Two-way ANOVA. 22 °C UT; $n = 4$, 22 °C ET; $n = 5$, 30 °C UT; $n = 5$, 30 °C ET; $n = 5$, paired 22 °C UT; $n = 5$, paired 22 °C ET; $n = 5$. Effect of ET within temperature; [**]$p < 0.01$. Effect of temperature within UT or ET; [###]$p < 0.001$. **e, f** The effect of housing temperature and ET at 22 °C and 30 °C on body fat (g) and lean body mass (g). Statistical testing; Two-way ANOVA (repeated measures). 22 °C UT; $n = 8$, 22 °C ET; $n = 10$, 30 °C UT; $n = 9$, 30 °C ET; $n = 10$. Effect of time within group; [$]$p < 0.05$, [$$]$p < 0.01$, [$$$]$p < 0.001$. Effect of ET within temperature; [*]$p < 0.05$, [***]$p < 0.001$. **g** The relative change in body fat (g) before to after the ET intervention in 22 °C, 30 °C, and paired 22 °C. Statistical testing; One-way ANOVA. $n = 10$. Effect between the groups as indicated with lines; [##]$p < 0.01$. (**h–j**) Effect of ET at 22 °C, 30 °C, and paired 22 °C on glucose tolerance. Statistical testing; Two-way ANOVA (repeated measures). 22 °C UT; $n = 8$, 22 °C ET; $n = 10$, 30 °C UT; $n = 9$, 30 °C ET; $n = 10$, paired 22 °C UT; $n = 10$, paired 22 °C ET; $n = 10$. Effect of ET on blood glucose response; [*]$p < 0.05$, [***]$p < 0.001$. Effect of ET on iAUC; [*]$p < 0.05$. (**k**) Effect of ET and temperature at 22 °C, 30 °C and paired 22 °C on glucose-stimulated insulin secretion at time 0 min and 20 min. Statistical testing; Two-way ANOVA (repeated measures). 22 °C UT; $n = 8$, 22 °C ET; $n = 10$, 30 °C UT; $n = 9$, 30 °C ET; $n = 10$, paired 22 °C UT; $n = 10$, paired 22 °C ET; $n = 10$. Effect of time within group; [$$]$p < 0.01$, [$$$]$p < 0.001$ Effect of temperature within UT or ET-groups (post); [##]$p < 0.01$, [###]$p < 0.001$. Effect of ET within temperature; [*]$p < 0.05$, [***]$p < 0.001$. Data are presented as mean ± SEM incl. individual values where applicable. The "$n = x$" defines the number of biologically independent animals used for the analyses except for panel d where $n = x$ defines the number of cages.

intake was 25% lower in both ET and UT mice housed at 30 °C, compared with mice housed at ambient temperature (Fig. 2d). Although the absolute food intake was lower in 30 °C, the relative increase induced by ET compared with the control mice was similar in both temperatures. No significant effect of ET on food intake was observed in the paired 22 °C ET mice (Fig. 2d).

Lean body mass increased similarly in all groups during the 6 weeks intervention, with no effect of housing temperature or ET (Fig. 2e). In UT mice, fat mass increased during the 6 weeks intervention (Fig. 2f). This fat mass gain was completely prevented in both 22 °C ET groups (Fig. 2f, g). In contrast, mice training at 30 °C only partially prevented this fat mass gain (Fig. 2f, g).

Glucose tolerance improves following voluntary wheel running ET in mice[31–34], but this has to the best of our knowledge only been tested for mice housed at 22 °C. Firstly, we observed a clear phenotype of housing temperature on fasting blood glucose, which was markedly lower at 30 °C housing independent of ET (Fig. 2h, i). In contrast to ET at 22 °C (Fig. 2h), ET at thermoneutral conditions did not improve glucose tolerance (Fig. 2i). This was likely due to reduced running volume in 30 °C housed mice, as glucose tolerance was also not improved in paired 22 °C ET (Fig. 2j). This was despite the reduction in fat mass at all temperatures (albeit less reduced at 30 °C ET) and similar improvements in running capacity by ET. As observed for the fasting blood glucose, we note that blood glucose curve was also noticeably lower during the glucose tolerance test (GTT) in UT 30 °C mice where it peaked at 12 mM compared to UT 22 °C mice where blood glucose peaked at 15 mM. That suggests that the 30 °C housing condition per se improves glucose tolerance in mice, which is in-line with a recent observation[34]. This is evident by comparing the UT mice from 22 °C and 30 °C housing, where there was a clear effect of housing temperature on glucose tolerance (Supplementary Fig. 2b), which is in agreement with the notion that the standard control mouse at 22 °C may be metabolically challenged[13]. In contrast to blood glucose, the 4-h fasted plasma insulin concentration was similar between all experimental groups (Fig. 2k). Glucose-stimulated plasma insulin was reduced by ET in both groups of 22 °C housed mice indicating improved insulin sensitivity, which was not observed in thermoneutrally housed ET mice (Fig. 2k). Interestingly, glucose-stimulated plasma insulin was 40% higher in mice housed at 30 °C compared to 22 °C (Fig. 2k), suggesting that housing temperature could affect β-cell function, however this was not

further explored in the current study. Housing temperature-induced metabolic changes extended to lipid metabolism, as fasting plasma triglyceride concentration was 25% higher in 30 °C housed mice compared to mice housed at 22 °C with no effect of ET (Supplementary Fig. 2c). In UT mice, fasting plasma free fatty acids were 135% higher at 30 °C than in 22 °C (Supplementary Fig. 2d).

Taken together, these findings show that exercise-induced improvements in body composition and glycemic control are increased at ambient housing temperature.

**Thermoneutral housing prevents the exercise-mediated improvement of insulin action independent of running volume.** We next investigated if housing temperature would affect the ET-induced adaptations on insulin action and glucose uptake in skeletal and cardiac muscle, and several adipose tissue depots.

Insulin was injected in the retro-orbital vein in anaesthetized mice. Insulin caused a drop of blood glucose that was similar between all experimental groups (Fig. 3a–c). Plasma insulin concentration was 1.8 ng/ml at tissue harvest, 10 min following the insulin injection, in all groups (Supplementary Fig. 3a).

Despite no apparent effect of ET on the change in whole-body blood glucose levels during insulin stimulation, insulin-stimulated glucose uptake was increased by ET in skeletal muscle (m. triceps brachii (triceps), +40%, Fig. 3d) at 22 °C. This effect was not observed following ET at 30 °C (Fig. 3d). This lack of ET-induced increase in muscle insulin-stimulated glucose uptake was ascribed to housing temperature, as the paired 22 °C ET group exhibited increased insulin-stimulated muscle glucose uptake (+35%, Fig. 3d). Similar results were observed in quadriceps muscle (Supplementary Fig. 3b). Neither housing temperature nor ET affected insulin-stimulated glucose uptake in the heart (Fig. 3e). As white (WAT) and brown (BAT) adipose tissue are responsive to temperature[35,36] as well as ET at ambient temperature[37,38], we next investigated if this also applied to ET in thermoneutral conditions. ET in 22 °C increased insulin-stimulated glucose uptake in inguinal (i)WAT, (+45%, Fig. 3f) and perigonadal (p) WAT (+70%, Fig. 3g), but not in BAT (Fig. 3h). This response was unaffected by thermoneutral housing in pWAT but blunted by 30 °C housing in iWAT. As for skeletal muscle, the differences in iWAT of ET-induced enhanced insulin action were ascribed to housing temperature rather than training volume as ET also enhanced insulin action in iWAT in paired 22 °C mice (Fig. 3f). 30 °C housing led to an 80% reduction in insulin-stimulated

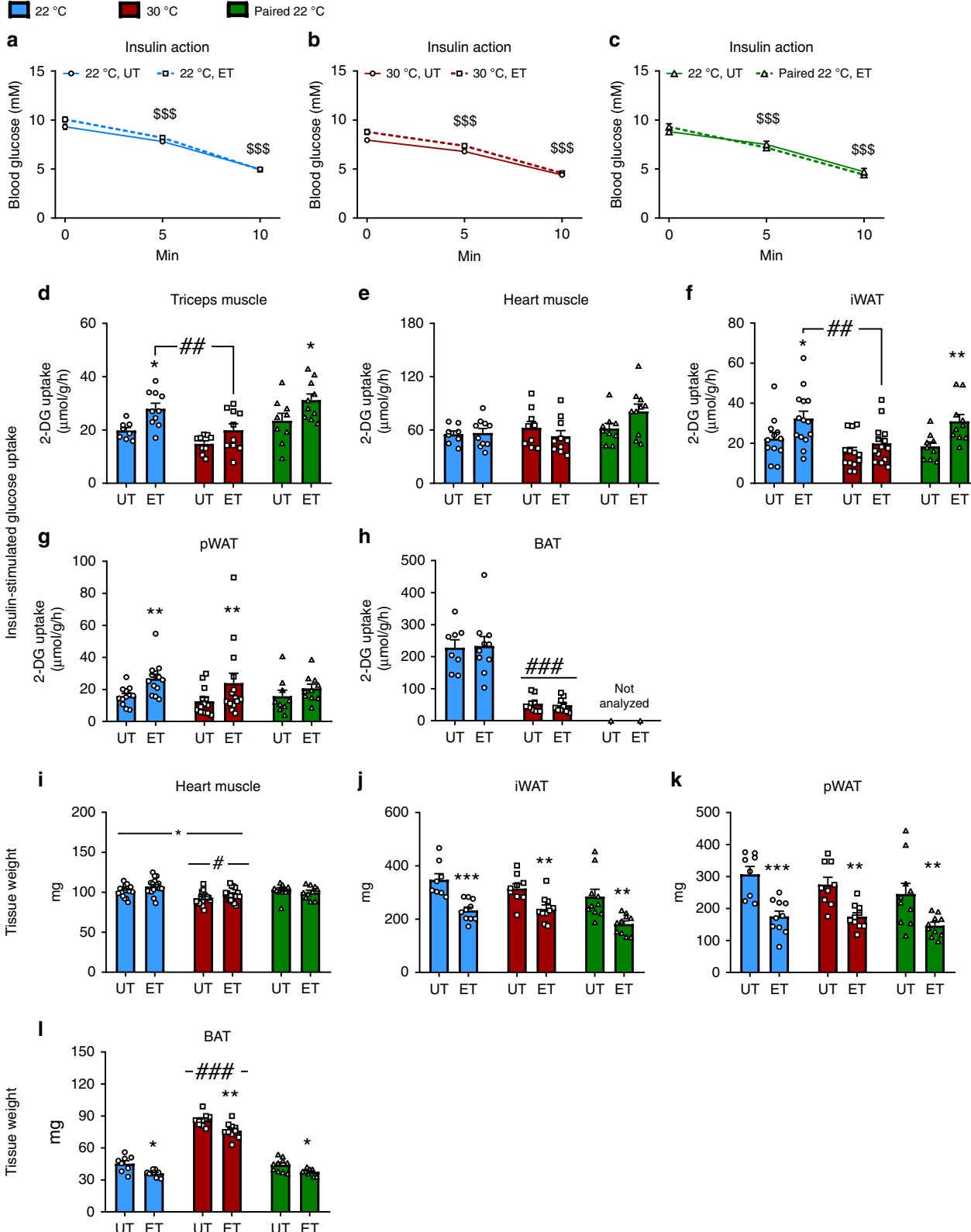

glucose uptake in BAT compared to 22 °C with no apparent effect of ET (Fig. 3h), and therefore the BAT from paired 22 °C ET mice was not analyzed. Thermoneutrality lowered basal glucose uptake in BAT by 85% with no effect observed in skeletal muscle or WAT depots (Supplementary Fig. 3c). ET did not alter basal glucose uptake in any of the analyzed tissues depots (Supplementary Fig. 3c). Importantly, 2-deoxy-glucose ($^3$H) tracer

activity in the blood was similar between UT and ET groups (Supplementary Fig. 3d).

We observed a similar cardiac hypertrophy after ET (+5%) at both housing temperatures and a 9.5% reduction in heart mass in 30 °C housed mice (Fig. 3i). However, the mass of the heart was unaffected by ET in the paired 22 °C ET group. The size of all analyzed fat depots was reduced similarly in all ET groups (iWAT; −30%,

**Fig. 3 Thermoneutral housing prevents the improved insulin action in skeletal muscle and adipose tissue following ET, independently of running volume. a–c** Effect of retro-orbital insulin injection (0.3U/kg) on blood glucose in 22 °C (**a**), 30 °C (**b**), and paired 22 °C (**c**), untrained (UT) and voluntary wheel running exercise-trained (ET). Statistical testing; Two-way ANOVA (repeated measures). 22 °C UT; $n = 12$, 22 °C ET; $n = 14$, 30 °C UT; $n = 13$, 30 °C ET; $n = 14$, paired 22 °C UT; $n = 10$, paired 22 °C ET; $n = 10$ mice. Effect of time (insulin); $^{\$\$\$}p < 0.001$. **d–h** Effect of ET on insulin-stimulated 2-deoxyglucose (2-DG) uptake in 22 °C, 30 °C, and paired 22 °C in skeletal muscle (m. triceps brachii) (**d**), cardiac muscle (**e**), iWAT (**f**), pWAT (**g**), and BAT (**h**). Statistical testing; Two-way ANOVA and student's *t*-test. 22 °C UT; $n = 8$ ($n = 12$ for iWAT and pWAT), 22 °C ET; $n = 10$ ($n = 14$ for iWAT and pWAT), 30 °C UT; $n = 9$ ($n = 13$ for iWAT and pWAT), 30 °C ET; $n = 10$ ($n = 14$ for iWAT and pWAT), paired 22 °C UT; $n = 10$, paired 22 °C ET; $n = 10$. Effect of ET within temperature; $^{*}p < 0.05$, $^{**}p < 0.01$. Effect of temperature as indicated with lines; $^{\#\#}p < 0.01$, $^{\#\#\#}p < 0.001$. **i** Effect of ET on cardiac muscle weight in 22 °C, 30 °C, and paired 22 °C (UT and ET groups). Statistical testing; Two-way ANOVA and student's *t*-test. 22 °C UT; $n = 16$, 22 °C ET; $n = 18$, 30 °C UT; $n = 16$, 30 °C ET; $n = 18$, paired 22 °C UT; $n = 10$, paired 22 °C ET; $n = 10$. Main-effect of ET; $^{*}p < 0.05$. Main-effect of temperature; $^{\#}p < 0.05$. **j–l** Effect of voluntary wheel running exercise training (ET) on fat depot weight (iWAT (**j**), pWAT (**k**), and BAT(**l**)) at 22 °C, 30 °C, and paired 22 °C, UT and ET respectively. Statistical testing; Two-way ANOVA and student's *t*-test. 22 °C UT; $n = 8$, 22 °C ET; $n = 10$, 30 °C UT; $n = 9$, 30 °C ET; $n = 10$, paired 22 °C UT; $n = 10$, paired 22 °C ET; $n = 10$. Effect of ET within temperature; $^{*}p < 0.05$, $^{**}p < 0.01$, $^{***}p < 0.001$. Main effect of temperature; $^{\#\#\#}p < 0.001$. Data are presented as mean ± SEM incl. individual values where applicable. The "n = x" defines the number of biologically independent animals used for the analyses.

Fig. 3j, pWAT; −40%, Fig. 3k, and BAT; −15%, Fig. 3l). iWAT (Fig. 3j) and pWAT (Fig. 3k) mass were unchanged by housing temperature, while BAT amount was doubled in thermoneutrally housed mice (Fig. 3l) supporting a recent study[20].

However, due to ET leading to decreased fat depot size, the total amount of glucose taken up by WAT-tissue in response to insulin was similar between UT and ET groups (Supplementary Fig. 3e). BAT still showed reduced glucose uptake at 30 °C housing temperature independent of ET (Supplementary Fig. 3e).

**Thermoneutrality alters the molecular adaptations to ET in skeletal muscle without affecting canonical insulin signaling.** Major molecular adaptations occur in skeletal muscle in response to ET, but such responses have, to the best of our knowledge, in mice only been studied at ambient temperature. We therefore determined the molecular responses to voluntary ET on known training responsive proteins. In triceps muscle, hexokinase (HK) II (Fig. 4a) and pyruvate dehydrogenase (PDH) (Fig. 4b) increased following ET only in 22 °C housed mice, thus housing temperature was a critical factor for induction of those proteins in response to training. Noticeably, the effect of ET with full access to the running wheel in 22 °C was greater on HKII expression compared to the paired training group (Fig. 4a). GLUT4, increased similarly (+20%, Fig. 4c, see 4D for representative blots) with voluntary ET irrespective of housing temperature, but not in the paired 22 °C ET mice (Fig. 4c), suggesting that at ambient temperature running volume determines ET-induced GLUT4 expression.

It is well known that mitochondrial content increases with ET. To our surprise, we observed a reduced response in four of the five complexes of the electron transport chain (ETC) following ET of 30 °C housed mice compared to 22 °C in triceps muscle (Fig. 4e–i, see 4j for representative blots). This effect was specifically due to housing temperature as in the paired 22 °C ET mice, all complexes increased after the intervention.

Some of these effects of housing temperature seemed to be muscle-type specific, as we did not observe obvious effects of housing temperature in quadriceps muscle (Supplementary Fig. 4a). We observed no effect of ET or temperature on protein expression of any of the above-described proteins in heart muscle (Supplementary Fig. 4b). No major changes in canonical insulin-stimulated insulin signaling molecules were observed with ET or housing temperature in any of the analyzed muscles (Supplementary Fig. 4c). The blunted response in insulin-stimulated glucose uptake in skeletal muscle at 30 °C following ET could therefore not be ascribed to altered intracellular insulin signaling, but rather changes in expression of glucose-handling proteins.

Collectively these data demonstrate that the ability of ET to increase insulin-stimulated glucose uptake and protein expression of key training responsive proteins in skeletal muscle was lost or diminished when the mice were housed at 30 °C, and this was not due to lower training volume.

**Adipose tissue molecular adaptations to ET in 22 °C are partially reduced by thermoneutral housing.** WAT and BAT adipose tissue are responsive to temperature changes[35,36] as well as ET at ambient temperature[37,38]. In iWAT, ET at 22 °C led to increased content of HK II, GLUT4, and PDH. These adaptations did not occur after ET in 30 °C (Fig. 5a). Like in muscle, glucose handling proteins are thus potentially involved in the mechanisms behind the observed differences in insulin-stimulated glucose uptake in WAT, as insulin signaling was unaltered (data not shown). HK II, GLUT4, and PDH also did not increase in the paired 22 °C ET mice in iWAT, suggesting that the lower running volume in 30 °C housed mice contributes to this difference. For BAT, HK II increased with ET in 22 °C and this effect was not observed at thermoneutrality. GLUT4 and PDH protein expression in BAT did not increase with ET in either temperatures. In accordance with the lower insulin-stimulated glucose uptake in BAT at 30 °C, we observed a generally lower HK II (−90%), GLUT4 (−20%), and PDH (−30%) protein expression (Fig. 5b).

ET increased protein content of four of five complexes of the ETC in iWAT in 22 °C, but not 30 °C housed mice (Fig. 5c). Paired 22 °C ET led to a significant increase only in complex III. Thus, both reduced running distance at thermoneutrality as well as temperature seem to underlie the differences in mitochondrial adaptation to ET in iWAT. UCP-1 protein expression in iWAT increased with ET at both housing temperatures (Fig. 5c). In BAT, only complex III increased with voluntary ET and this occurred at both housing temperatures (Fig. 5d). Thermoneutral housing reduced complex I and II, as well as UCP-1 protein in BAT compared with 22 °C (Fig. 5d). With strikingly no or only little effect observed of ET in BAT at either temperature, and therefore likely not a key target for the observed phenotype in 30 °C-housed mice, paired 22 °C ET BAT was not investigated. Representative blots are shown in Fig. 5e.

We also analyzed gene expression of proteins involved in thermogenesis and mitochondrial uncoupling in iWAT and BAT. As expected, gene expression of proteins involved in thermogenesis (*Ucp1, Cidea, Prdm16,* and *PGC-1α*) were all downregulated by thermoneutral housing in all depots investigated (Supplementary Fig. 5a, b). These genes were largely unaffected by ET in both temperatures, likely due to the fact that the running wheels were locked for 24 h prior to tissue dissection.

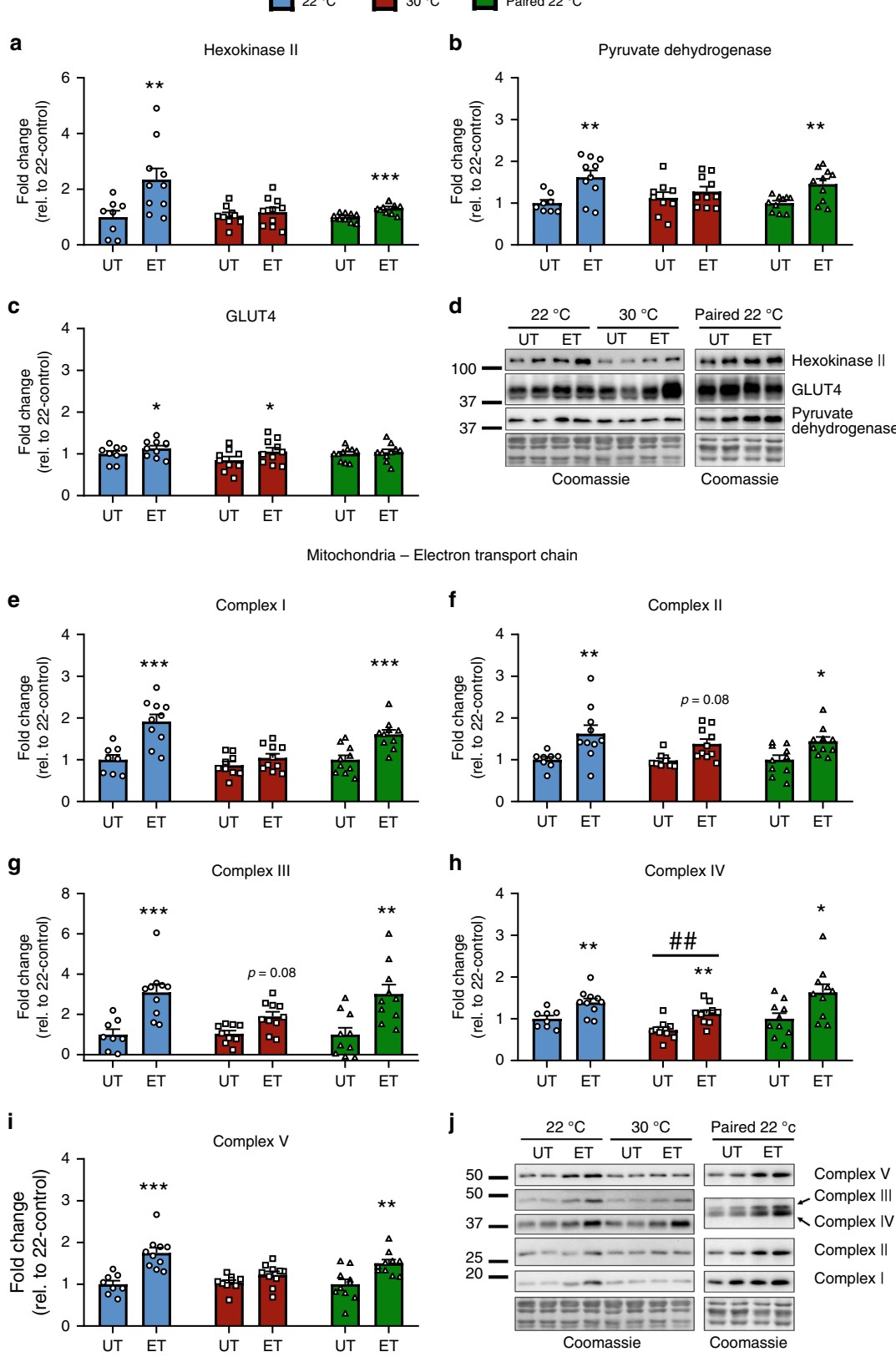

Taken together, we observed diminished molecular training adaptations in iWAT of thermoneutrally housed mice emphasizing the importance of considering housing temperature when performing ET studies investigating fat depots and metabolism.

**Thermoneutral housing supersedes the effect of exercise on gut microbiome composition**. Metabolic health has during recent years been shown to be under strong influence by the gut microbiome[39]. We therefore tested if housing temperature affects ET-induced gut microbiome adaptations.

**Fig. 4 Thermoneutrality alters the molecular adaptations to ET in skeletal muscle without affecting canonical insulin signaling. a–c** Effect of exercise training (ET) at 22 °C, 30 °C, and paired 22 °C on training responsive proteins in triceps muscle. Representative blots are shown in (**d**). Statistical testing; Two-way ANOVA and student's *t*-test. 22 °C untrained (UT); $n = 8$, 22 °C ET; $n = 10$, 30 °C UT; $n = 9$, 30 °C ET; $n = 10$, paired 22 °C UT; $n = 10$, paired 22 °C ET; $n = 10$. Effect of ET within temperature; *$p < 0.05$, **$p < 0.01$, ***$p < 0.001$. **e–i** Effect of ET at 22 °C, 30 °C, and paired 22 °C on subunits of the electron transport chain of the mitochondrion in triceps muscle. Representative blots are shown in (**j**). Statistical testing; Two-way ANOVA and student's *t*-test. 22 °C UT; $n = 8$, 22 °C ET; $n = 10$, 30 °C UT; $n = 9$, 30 °C ET; $n = 10$, paired 22 °C UT; $n = 10$, paired 22 °C ET; $n = 10$. For *p*-values described, a Two-way ANOVA was applied. Effect of ET within temperature; *$p < 0.05$, **$p < 0.01$, ***$p < 0.001$. Effect of temperature within UT or ET groups; ##$p < 0.01$. Data are presented as mean ± SEM incl. individual values. The "$n = x$" defines the number of biologically independent animals used for the analyses.

Gut microbiome clustering following the different interventions is shown in Fig. 6a. After adjusting for the conditional effect of time on the experimental units, both UT and ET mice showed significant differences in gut microbiome composition with distinct separation depending on housing temperature (bottom panel of Fig. 6a, $r > 0.13$, $p < 0.04$). Differences in individual group-clustering are shown in Supplementary Fig. 5c. The largest effect on the abundance of specific phylotypes was observed between the UT mice that were housed at either 22 or 30 °C. A total of 32 phylotypes showed increased abundance when housing mice in 30 °C compared to 22 °C (Fig. 6b). In particular, four phylotypes (zOTU_572, zOTU_534, zOTU_655, zOTU_593) of the family *Lachnospiraceae* stood out as having substantially higher relative abundance compared to other phylotypes in 30 °C housing.

With only five phylotypes changed with ET at 30 °C and four changed in 22 °C, the effect of housing temperature superseded the effect of ET on the gut microbiome composition. At 22 °C, ET reduced the abundance of four phylotypes (Fig. 6c), while ET decreased four and increased one phylotype in 30 °C (Fig. 6d). Importantly, the decrease relative abundance of four phylotypes (zOTU_95, zOTU_81, zOTU_74, zOTU_56) were the same in both temperatures, suggesting these phylotypes of the family *Muribaculaceae* are conserved exercise-responsive bacteria independent of housing temperature (Fig. 6c, d). These four phylotypes did not change in the paired training group (Fig. 6e).

The two dominant phyla in the murine gut microbiome are *Bacteroidetes* and *Firmicutes*[40], as also observed in the current study (Fig. 6f). The ratio between these species have been suggested to be altered in obesity[41–43] and aging[44]. Despite the changes in body composition, glycemic control, and exercise capacity observed in the current study, we did not see any effects of housing temperature or ET in the abundance of *Bacteroidetes* (Fig. 6g), *Firmicutes* (Fig. 6h), or the ratio between the two (Fig. 6i).

The above data demonstrate that thermoneutral housing temperature per se causes a remarkable modification of the gut microbiome composition in mice with modest effects of ET.

## Discussion

Our major finding was that thermoneutral housing temperature blunts or even abolishes systemic metabolic as well as molecular adaptations to voluntary wheel running ET in mice. Hence, several previously reported effects of ET could be secondary to the metabolic stress that mice experience at 22 °C. We thus identify housing temperature as a critical factor for exercise adaptations in mice (Supplementary Fig. 6).

Most remarkably, the ET response on glucose metabolism was reduced by thermoneutral housing. ET at ambient temperature led to a robust increase in glucose tolerance and improved insulin-stimulated glucose uptake in skeletal muscle and WAT in agreement with many previous reports[45–53]. However, this effect was absent in thermoneutrally housed mice. With regards to glucose tolerance, this could be ascribed to reduced running volume in thermoneutrally housed mice as paired-trained 22 °C

mice, which ran the same distance as the 30 °C mice, also did not improve glucose tolerance with training. In contrast to glucose tolerance, the blunted ET-induced enhanced insulin-stimulated glucose uptake observed in 30 °C housed mice, could be solely ascribed to housing temperature and not training volume since the paired 22 °C mice improved insulin sensitivity just as 22 °C mice when compared to adaptations in 22 °C. In addition, while 22 °C-housing led to mild cold stress, the observed changes (e.g., running distance) were not due to overheating of 30 °C-housed mice.

Our finding that the voluntary wheel running model is less efficient in improving metabolic status in mice housed at 30 °C, on the one hand might indicate that thermoneutral housing is not a good experimental choice when investigating molecular or metabolic events underlying the benefits of exercise. On the other hand, it could indicate that several of the previously reported effects of training in mice are secondary to the mild cold stress that mice experience at 22 °C. In agreement with the latter, the minimal ET effects in our young lean mice at 30 °C on glucose tolerance and insulin-stimulated glucose uptake, mimics the human condition well. While ET repeatedly has been documented to improve glucose tolerance in obese and type 2 diabetic subjects[54–56], ET often does not result in improved glucose tolerance in lean healthy subjects[55,57,58]. Notably, at ambient housing temperature over one-third of total energy expenditure in mice is cold-induced thermogenesis[59]. In contrast, cold-induced thermogenesis contributes a very small fraction to total energy expenditure in humans[60]. Increasing housing temperature from ambient temperature to 27 °C–30 °C improves the metabolic similarity between humans and mice and has been suggested to be a better housing strategy[9,10,27,28,30]. Thus, it is possible that the improvements in glucose tolerance often reported in mice housed at ambient temperature reflects an experimental artefact caused by chronic mild cold stress. In support of this, mice housed at ambient temperature displayed glucose intolerance compared with mice housed at 30 °C. Indeed, the control mouse housed at ambient temperature has been argued to be a control that is poor for human standards[9,13].

At 22 °C, the mouse also has an extraordinarily high running volume, which, combined with mild cold stress could intensify the effects of ET in this model. Furthermore, the maximal oxygen uptake (ml/kg/min) is already substantially higher in UT mice compared to humans (including trained individuals)[61,62]. All of the above characteristics could have implications for the translatability to the human condition when investigating training interventions in a genetic model or testing pharmacological compounds. This has indeed been observed for many other molecular mechanisms, where conclusions drawn from mice housed at 22 °C have been completely changed when investigated at thermoneutrality[17,19,21,22,24,25,63,64]. Because ET improves glucose tolerance in obese and type 2 diabetic human subjects[54–56], it would be relevant for future studies to investigate voluntary ET in thermoneutrality in diet-induced or diabetic insulin resistant mice.

Another major finding of our study was that the molecular adaptations of key exercise-responsive proteins were blunted or

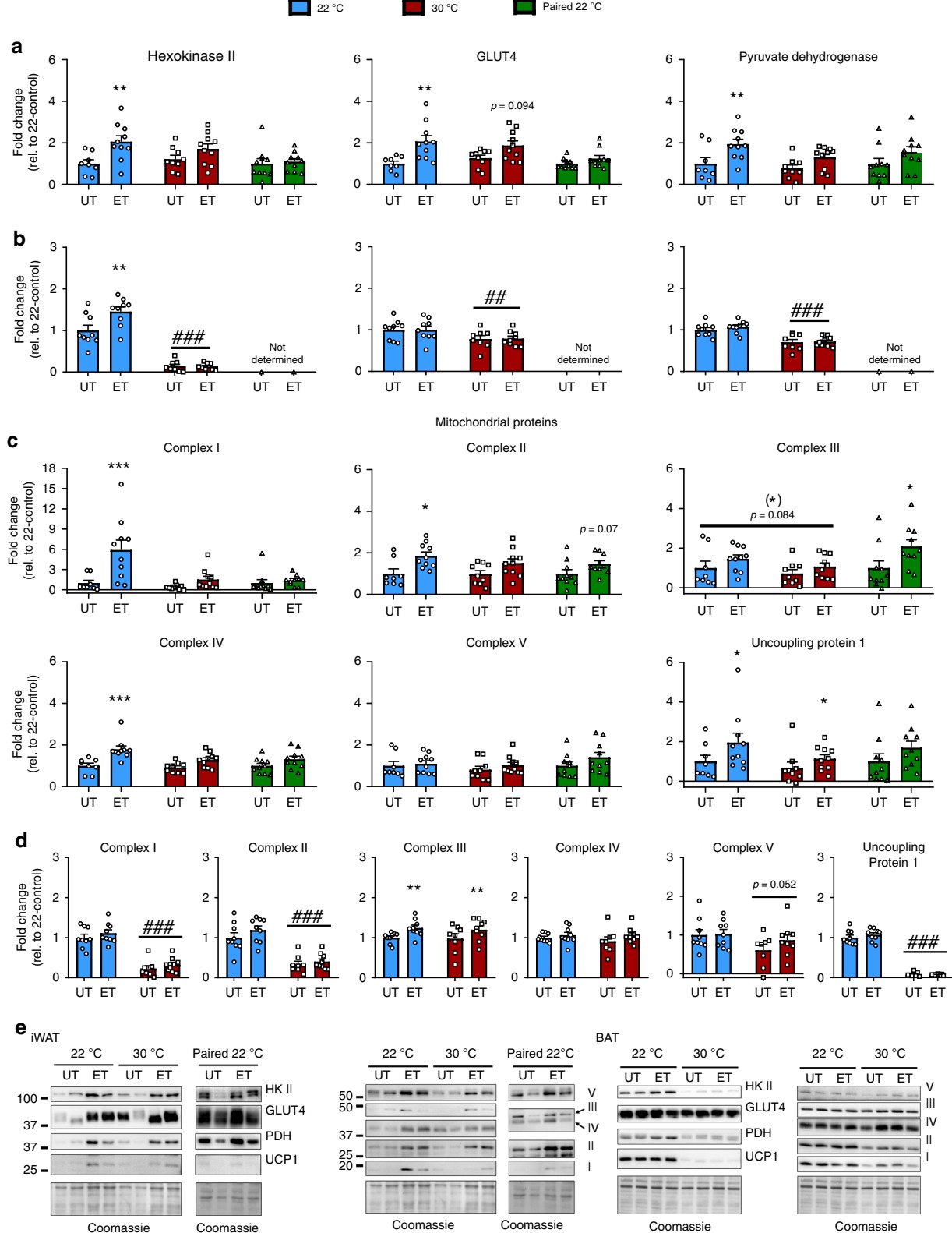

even lost at thermoneutrality, independently of training volume. In triceps muscle, hexokinase II (upholds the glucose gradient across the membrane by phosphorylating entering glucose) and PDH (converts pyruvate to acetyl-CoA connecting glycolysis and the Krebs cycle) were only significantly upregulated by ET in ambient temperature, not thermoneutrality. This was also

apparent for subunits of the ETC in the mitochondria. Interestingly, these differences in mitochondrial adaptations in muscle and fat depots were not reflected in differences in improvements in exercise performance, as running capacity increased equally in all ET groups. This resembles data from a recent study in male mice, where endurance capacity was not different between mice

**Fig. 5 Adipose tissue molecular adaptations to exercise training are partially reduced by thermoneutral housing. a** Effect of exercise training (ET) at 22 °C, 30 °C, and paired 22 °C on training responsive proteins in iWAT. Representative blots are shown in (**e**). Statistical testing; Two-way ANOVA and student's t-test. 22 °C untrained (UT); n = 8, 22 °C ET; n = 10, 30 °C UT; n = 9, 30 °C ET; n = 10, paired 22 °C UT; n = 10, paired 22 °C ET; n = 10. For p-values described, a two-way ANOVA was applied. Effect of ET within temperature; **p < 0.01. **b** Effect of ET at 22 °C and 30 °C on training responsive proteins in BAT. Representative blots are shown in (**e**). Statistical testing; Two-way ANOVA. 22 °C UT; n = 9, 22 °C ET; n = 9, 30 °C UT; n = 8, 30 °C ET; n = 9. Effect of ET within temperature; **p < 0.01. Effect of temperature within UT or ET groups; ##p < 0.01, ###p < 0.001. **c** Effect of ET at 22 °C, 30 °C, and paired 22 °C on mitochondrial proteins in iWAT. Representative blots are shown in (**e**). Statistical testing; Two-way ANOVA and student's t-test. 22 °C UT; n = 8, 22 °C ET; n = 10, 30 °C UT; n = 9, 30 °C ET; n = 10, paired 22 °C UT; n = 10, paired 22 °C ET; n = 10. Effect of ET within temperature; *p < 0.05, ***p < 0.001. **d** Effect of ET at 22 °C and 30 °C on mitochondrial proteins in BAT. Representative blots are shown in (**e**). Statistical testing; Two-way ANOVA. 22 °C UT; n = 9, 22 °C ET; n = 9, 30 °C UT; n = 8, 30 °C ET; n = 9. Effect of ET within temperature; **p < 0.01. Effect of temperature within UT or ET groups; ###p < 0.001. Data are presented as mean ± SEM incl. individual values. The "n = x" defines the number of biologically independent animals used for the analyses.

trained at 29 °C compared to mice trained at 22 °C[34]. Collectively, this indicates that thermoneutral housing does not affect exercise capacity in mice.

Metabolic training-induced improvements are often associated with reduced adiposity[65,66]. In our study, ET reduced body-fat at both housing temperatures, although to a lesser extent at thermoneutrality. Interestingly, our paired running group elucidated that this difference was a cause of housing temperature and not training volume. A better effect of ET on reducing adiposity at 22 °C is likely due to the higher metabolic demand on mice at 22 °C that, because of mild cold stress, exhibit increased energy expenditure as has been described previously in UT mice[67]. Our metabolic measurements of mice during voluntary wheel running at different temperatures support this, as mice running at ambient temperature displayed 60% higher energy expenditure which in part is due to the higher running volume. This could be a contributing factor for a lower body fat percentage seen in trained mice at 22 °C compared to 30 °C. When showing the energy usage per wheel turn in running mice, Virtue and colleagues (2012) showed that mice housed at 28 °C have a higher energy usage per wheel turn compared to 21 °C housed mice[68]. This parameter was not investigated in the current study, however, in the study by Virtue and colleagues, the mice were not habituated to thermoneutral temperatures prior to the experiment, which makes a direct comparison challenging.

In addition to a reduced metabolic rate in the current investigation, mice housed at 30 °C did not increase their RER during the dark cycle. A lower RER is indicative of higher relative contribution of fat as energy substrate. Taken together with the observed higher FFA and TG levels of 30 °C-housed mice (both in agreement[20] and disagreement with recent reports[65]), the data from the current study shows that fat metabolism might also be highly affected by housing temperatures and warrants further investigations.

Although not a key objective of the study, we found generic differences between housing temperatures in our UT mice that to the best of our knowledge, have not previously been documented. An important finding in our study was that the mouse gut microbiome was remarkably affected by housing temperature, but with only minimal effect of ET. The minimal effect on the gut microbiome is in contrast to a recent report, where marathon runners had increased abundance of *Veillionella* spp. post-marathon running and transfer of this bacterium to mice significantly increased their treadmill performance[69]. The effect of temperature on the gut microbiome is important as it has been shown to affect several functions in physiology, e.g. glucose metabolism[39,41,70], and has recently been associated with muscle function[71]. This effect of housing temperature alone has not to our knowledge been clearly demonstrated previously.

We also observed that chronic housing mice at 30 °C led to increased nightly activity. In contrast, acute changes in temperature did not change nightly activity, in agreement with another study[72]. Collectively, these observations indicate that appropriate acclimatization time is needed when investigating the effect of housing temperature on activity levels in mice.

Increasing housing temperature also led to a ~10% reduction in heart mass, suggesting that ambient housing leads to cardiac hypertrophy, likely due to a higher cardiac stress as indicated by twice as high heart rate at 22 °C (600 bpm) compared to 30 °C (300 bpm) housed mice[15,73,74].

We also observed a lower fasting blood glucose of thermoneutrally housed mice in agreement with another report[19]. Whether this is a direct effect of housing temperature, or due to the fact that mice housed at thermoneutrality ate less is not clear. In addition, we observed that thermoneutral housing led to elevated insulin secretion following a glucose challenge. Based on our findings, housing temperature may affect the outcome of studies investigating all of these processes, the interpretation of the results, and ultimately the translation to humans.

Contemporary biomedical research is using unbiased 'omics' approaches to comprehensively explore the global regulations to map the beneficial changes that occur with ET[51,75–77]. Such studies will need to be followed up by hypothesis-driven research genetically manipulating or pharmacologically inhibiting/activating a pathway of interest in order to elucidate the mechanistic role for a given ET-regulated protein or process. Considering the optimal housing condition for such studies might increase the translatability and clinical relevance for humans.

In conclusion, numerous training adaptations are reduced or even abolished by thermoneutral housing; the majority of which was not ascribed to a lower voluntary running volume in mice housed at thermoneutrality compared to mice housed at 22 °C. Thus, some reported effects of ET may be secondary to the combined effect of ET and the metabolic stress that mice experience at 22 °C. Our findings highlight that organismal and molecular adaptations to ET in mice depend upon housing temperature and that housing temperature is important to consider when using mice as an experimental model.

## Methods

**Animals.** Ten-week-old female C57BL/6J mice (Taconic, Lille Skensved, Denmark) were maintained on a 12:12-h light-dark cycle and received standard rodent chow diet (Altromin no. 1324; Chr. Pedersen, Denmark) and water ad libitum with nesting materials. We have complied with all relevant ethical regulations for animal testing and research. All experiments were approved by the Danish Animal Experimental Inspectorate (Licence; 2016-15-0201-01043). Mice were randomly assigned to ambient temperature (22 °C ± 1 °C) or thermoneutrality (30 °C ± 1 °C) in different rooms in the same animal facility. After a 7–10 day acclimatization period, mice were pair-housed and housed with or without free access to running wheels for 6 weeks. Pair-housed mice often run simultaneously and thus the distance recorded weekly for each cage does not reflect the running distance of each mouse. Core temperature was measured with a rectal thermometer at 1:00 pm (light period) and 9:00 pm (most active dark period). For paired ET mice were

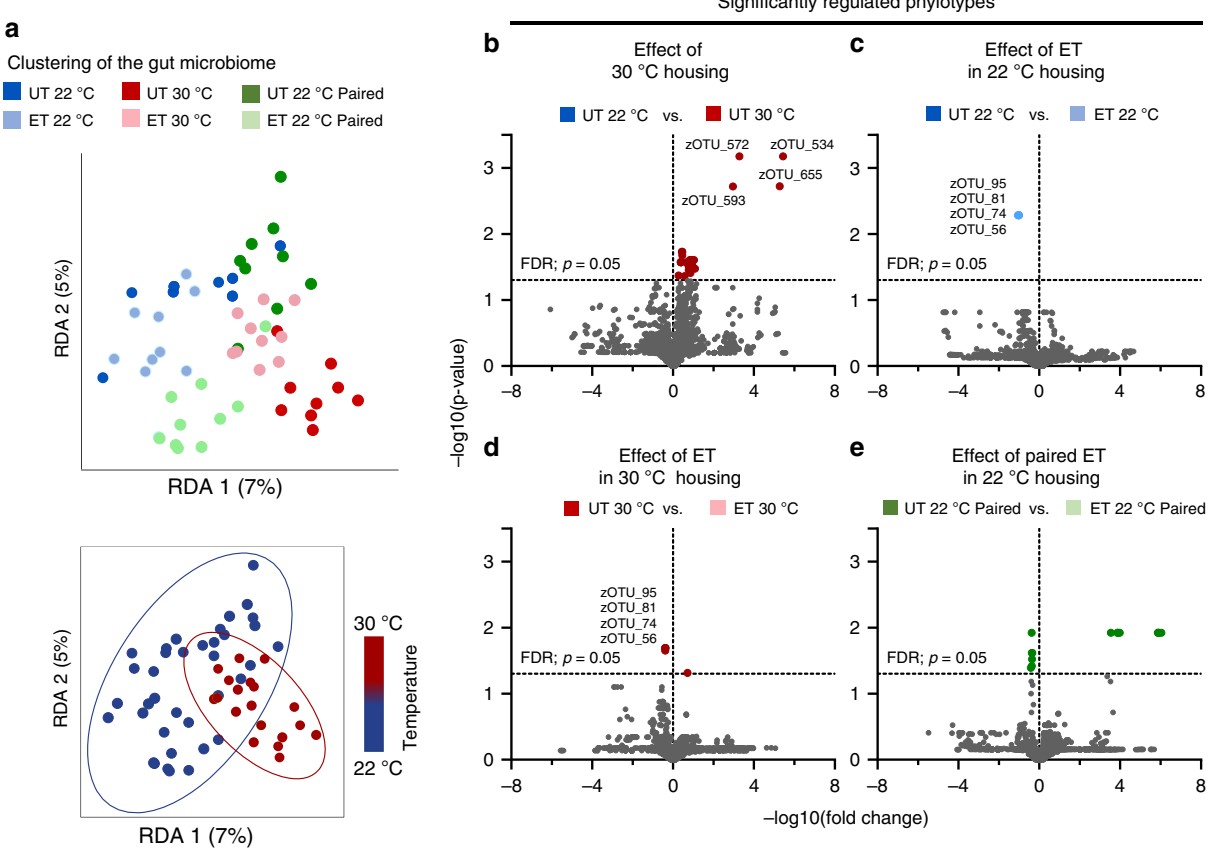

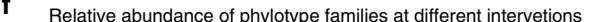

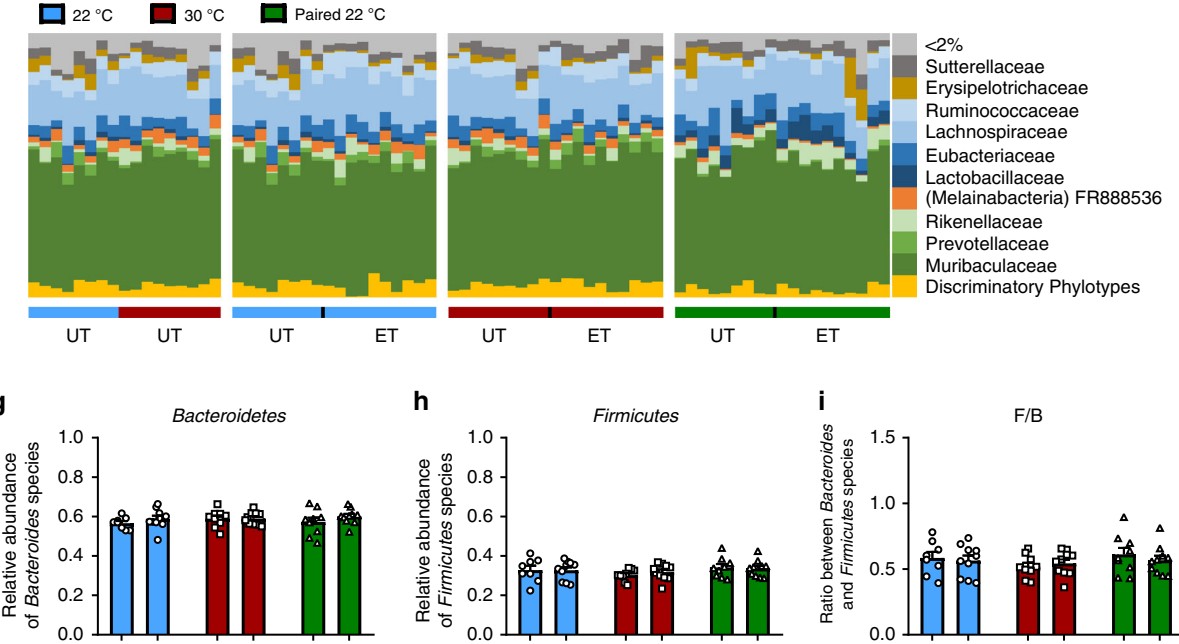

housed at ambient temperature with wheels that were locked from 00:00am to 5:00 pm. Running wheels were locked 24 h before GTS and terminal procedures to avoid any residual effects of acute exercise.

**Maximal running capacity.** Mice were acclimated to the treadmill three times (10 min at 0.16 m/s) within a week prior to the maximal running tests. The

maximal running test started at 0.16 m/s for 300 s with 15° incline, followed by a continuous increase (0.2 m/s) in running speed every 60 s until exhaustion (Treadmill TSE Systems, Germany). All running capacity tests were performed at 22 °C in a separate room from the housing rooms. This was done to be able to randomize the experimental groups, as well as being able to perform this test blinded (one person placing the mice on the treadmills, another person (blinded) to determine the exhaustion of the individual mouse).

**Fig. 6 Thermoneutral housing supersedes the effect of exercise on gut microbiome composition. a** Housing mice at 30 °C markedly alters gut microbiome composition with minor effect of exercise training (ET). Upper panel: dbRDA on Bray-Curtis distance of cecal 16S rRNA gene (V3-region) amplicons (zOTU level) and adjusted for the time effect of the paired-treatments (See Supplementary Fig. 5c for unconstrained Bray-Curtis PCoA plot). Lower panel: dbRDA plot of cecal (similar as above) clustering GM samples by housing temperature. The groups are separated by the colors indicated in the figure. ANOSIM test determined significant differences between experimental groups with mice housed at 22 °C, 30 °C, and paired 22 °C [$R > 0.13$, $p < 0.04$], were observed. 22 °C untrained (UT); $n = 8$, 22 °C ET; $n = 10$, 30 °C UT; $n = 9$, 30 °C ET; $n = 10$, paired 22 °C UT; $n = 10$, paired 22 °C ET; $n = 10$. **b–e** Volcano plots showing the change of specific phylotypes (zOTUs) within the different experimental groups; untrained mice at 22 °C and 30 °C (**b**), ET at 22 °C (**c**), ET at 30 °C, and ET at paired 22 °C; $t$-test, FDR $p \leq 0.05$ indicated. 22 °C UT; $n = 8$, 22 °C ET; $n = 10$, 30 °C UT; $n = 9$, 30 °C ET; $n = 10$, paired 22 °C UT; $n = 10$, paired 22 °C ET; $n = 10$. **f** The relative abundance of species summarized to the family level as indicated. 22 °C UT; $n = 8$, 22 °C ET; $n = 10$, 30 °C UT; $n = 9$, 30 °C ET; $n = 10$, paired 22 °C UT; $n = 10$, paired 22 °C ET; $n = 10$. **g–i** The relative abundance of *bacteroidetes* (**g**) and *Firmicutes* (**h**), and the ratio (**i**) between these two phyla. Statistical testing; Two-way ANOVA. 22 °C UT; $n = 8$, 22 °C ET; $n = 10$, 30 °C UT; $n = 9$, 30 °C ET; $n = 10$, paired 22 °C UT; $n = 10$, paired 22 °C ET; $n = 10$. The "n = x" defines the number of biologically independent animals used for the analyses.

### Table 1 Antibodies.

| Antibody | Source | Dilution | Catalog number (#) |
|---|---|---|---|
| Akt1/2 Ser473/474 | Cell Signaling Technology | 1:1000, 2% skim milk | 4051 |
| Akt2 | Cell Signaling Technology | 1:1000, 2% skim milk | 3063 |
| TBC1D4 Thr642 | Cell Signaling Technology | 1:1000, 2% skim milk | 4288 |
| HK II | Cell Signaling Technology | 1:1000, 2% skim milk | 2867 |
| GLUT4 | Thermo Fisher Scientific | 1:1000, 2% skim milk | PA1-1065 |
| TBC1D4 | Abcam | 1:1000, 2% skim milk | Ab189890 |
| OXPHOS | Abcam | 1:500, 2% skim milk | Ab110413 |
| UCP1 | Abcam | 1:1000, 2% skim milk | Ab10983 |
| Pyruvate Dehydrogenase | Grahame Hardie, University of Dundee, UK | 1:1000, 2% skim milk | |

**Body composition**. Total, fat and lean body mass were measured weekly by nuclear magnetic resonance using an EchoMRI™ (USA).

**Metabolic chambers**. After a 3-day acclimation period in the metabolic cages, oxygen consumption, ambulant activity (beam breaks), food intake and running distance/speed were measured by indirect calorimetry in a CaloSys apparatus during at least 2 days (TSE Systems, Bad Homburg, Germany). To test the acute effects of thermoneutrality, a group of mice were housed in the metabolic chambers at 22 °C for 3 days followed by an increase in the temperature to 30 °C for 3 days. All mice were single-housed during housing in metabolic cages.

**Glucose tolerance test**. Glucose (2.0 g/kg) was intraperitoneally injected into 5-h-fasted (fasting from 7:00am in single cages) mice in either 22 °C or 30 °C. Blood was collected from the tail vein at time points 0, 20, 40, 60, and 90 min and analyzed for glucose using a glucometer (Bayer Contour; Bayer, Münchenbuchsee, Switzerland). At time point 0 and 20 min, insulin was analyzed in duplicates in plasma (#80-INSTRU-E10; ALPCO Diagnostics).

**In vivo insulin-stimulated ³H-2-DG uptake**. To determine 2-deoxyglucose (2-DG) uptake in muscle, [³H]2-DG (Perkin Elmer) was injected retro-orbitally in a bolus of saline containing 66.7 µCi/mL [³H]2DG corresponding to ~9–10 µCi/mouse (6 µL/g body weight) in chow. The injectate also contained 0.3 U/kg body weight insulin (Actrapid; Novo Nordisk, Bagsværd, Denmark) or a comparable volume of saline. Prior to stimulation, mice were fasted for 3 h from 07:00am and anaesthetized (intraperitoneal injection of 7.5 mg pentobarbital sodium/100 g body weight) for 15 min. Blood samples were collected from the tail vein immediately prior to insulin or saline injection and after 5 and 10 min and analyzed for glucose concentration using a glucometer (Bayer Contour; Bayer, Münchenbuchsee, Switzerland). After 10 min, all tissues were excised, weighed (fat depots only), and quickly frozen in liquid nitrogen and stored at −80 °C until processing. Blood was collected by punctuation of the heart, centrifuged and plasma frozen at −80 °C. Plasma samples were analyzed for insulin concentration and specific [³H]2DG tracer activity. Tissue-specific 2DG uptake was analyzed as described elsewhere[78,79]. Here, muscle samples were weighed and homogenized in 0.5% perchloric acid. Homogenates were centrifuged and neutralized with KOH. One aliquot was counted directly to determine [³H]2DG and [³H]2DG-6-phosphate radioactivity. A second aliquot was treated with Ba(OH)₂ [0.1 M] and ZnSO₄ [0.1 M] to remove [³H]2DG-6-phosphate and any tracer incorporated into the glycogen and then counted to determine [³H]2DG radioactivity. The uptake-specific [³H]2DG-6-phosphate is the difference between the two aliquots. The principle behind this method is that the [³H]2DG, delivered retro-orbitally, is taken up by the tissues. When homogenized, the tissue lysate contains both [³H]2DG-6-phosphate and [³H]2DG. As only [³H]2DG-6-phosphate is trapped inside the myofibers because muscle does not express G-6-phosphatase, relating the lysate

content of [³H]2DG-6-phosphate to [³H]2DG isolates the glucose actually transported into the myofibers. Furthermore, relating the amount of [³H]2DG -6-phosphate taken up by the myofiber with available [³H]2DG (plasma tracer availability, presented in Supplementary Fig. 3d), the specific amount of glucose entering the muscle cells was estimated.

**Plasma analyses**. Plasma insulin concentration was analyzed in duplicates (#80-INSTRU-E10; ALPCO Diagnostics). Plasma triacylglyceride (TG) was analyzed in duplicates (#Triglycerides CP, Horiba ABX). Plasma free fatty acids (FFA) were analyzed in duplicates (#NEFA C ACS-ACOD, Wako Chemicals).

**Tissue processing**. Muscles were pulverized in liquid nitrogen. Muscle and adipose tissue were homogenized 2 × 0.5 min at 30 Hz using a TissueLyser II bead mill (Qiagen, USA) in ice-cold homogenization buffer (10% glycerol, 1% NP-40, 20 mM sodium pyrophosphate, 150 mM NaCl, 50 mM HEPES (pH 7.5), 20 mM β-gly-cerophosphate, 10 mM NaF, 2 mM phenylmethylsulfonyl fluoride (PMSF), 1 mM EDTA (pH 8.0), 1 mM EGTA (pH 8.0), 2 mM Na3VO4, 10 µg/mL leupeptin, 10 µg/mL aprotinin, 3 mM benzamidine). Following end-over-end rotation for 30 min at 4 °C, the samples were centrifuged (9500 × g) for 20 min at 4 °C. Thereafter, the supernatants (clear lysate) were collected and stored at −80 °C. The latter two steps (centrifugation and lysate collection) were performed three times in adipose tissue to avoid contamination of fatty acids.

**Immunoblotting**. Lysate protein concentrations were measured using the bicinchoninic acid (BCA) method with bovine serum albumin (BSA) as standard. Total protein and phosphorylation levels of relevant proteins were determined by standard immunoblotting techniques loading equal amounts of protein. The primary antibodies used are presented in Table 1. Polyvinylidene difluoride membranes (Immobilon Transfer Membrane; Millipore) were blocked in Tris-buffered saline (TBS)-Tween 20 containing 2% milk protein for 5 min at room temperature. Membranes were incubated with primary antibodies overnight at 4 °C, followed by incubation with horseradish peroxidase-conjugated secondary antibody for 45 min at room temperature. Coomassie brilliant blue staining was used as a loading control[80]. Bands were visualized using the Bio-Rad ChemiDoc MP Imaging System and enhanced chemiluminescence (ECL+; Amersham Biosciences).

**qPCR analyses**. Total RNA was extracted from BAT, iWAT, and pWAT depots using TRI reagent (T9424, Sigma-Aldrich) followed by isolation using RNeasy Mini Kit (74106, Qiagen). Reverse transcription was carried out on 1000 ng RNA using the High Capacity cDNA Reverse Transcription kit (4368814, Applied Biosystems). Gene expression was determined based on real-time quantitative PCR using SYBR green (PP00259, Primerdesign). The data was analyzed with the ΔΔCT method and normalized to the housekeeping gene 36b4. All primers are listed in Table 2.

**Table 2 Primer.**

| Gene | Forward primer (5′→3′) | Reverse primer (5′→3′) |
|------|------------------------|------------------------|
| Ucp1 | GGATTGGCCTCTACGACTCA | TAAGCCGGCTGAGATCTTGT |
| pan-Pgc1a | TGATGTGAATGACTTGGATACAGACA | GCTCATTGTTGTACTGGTTGGATATG |
| Prdm16 | CCTGTGGGAGTCCTGAAAGA | CAGCTTCTCCGTCATGGTTT |
| Cidea | GTCAAAGCCACGATGTACGA | CAGGAACTGTCCCGTCATCT |

**Microbiota analysis (samples collection, processing, and DNA extraction).** Caecum fecal samples were collected from sedated mice during the terminal experiment with retro-orbital injections. Approximately 200 mg of the caecal content were used for DNA extraction using the PowerSoil® DNA Isolation Kit (MOBIO Laboratories, Carlsbad, CA, USA), following the instructions of the manufacturer, but with minor modifications. Briefly, prior to DNA extraction, samples were placed into the PowerBead tubes and heat-treated at 65 °C for 10 min and then at 95 °C for 10 min. Subsequently, solution C1 was added and bead-beating performed in FastPrep (MP Biomedicals, Santa Ana, CA, USA) using three cycles of 15 s each, at a speed of 6.5 m s$^{-1}$. The remaining DNA extraction procedure followed the manufacturer's instructions.

**High-throughput 16S rRNA gene amplicon sequencing.** Gut microbiome composition was determined by high-throughput 16S rRNA gene amplicon sequencing. The primers designed with adapters Nextera Index Kit® (Illumina, CA, USA) targeted the V3 region (~190 bp) and the library preparation, purification and sequencing were performed as previously described[81]. Briefly, the amplification profile (1st PCR) followed: Denaturation at 95 °C for 2 min; 33 cycles of 95 °C for 15 s, 55 °C for 15 s and 68 °C for 40 s; followed by final elongation at 68 °C for 5 min, while barcoding (2nd PCR) was performed at 98 °C for 1 min; 12 cycles of 98 °C for 10 s, 55 °C for 20 s and 72 °C for 20 s; elongation at 72 °C for 5 min. The amplified fragments with adapters and tags were purified and normalized using custom made beads, pooled and subjected to 150 bp pair-ended NextSeq (Illumina, CA, USA) sequencing.

Sequencing of the 16S rRNA gene (V3-region) amplicons yielded 3,434,893 high-quality reads (mean sequence length of 183 bp) and the number of reads per sequenced sample varied from 50,140 to 141,905 with an average of 85,872 (SD 20,246).

**Processing of high throughput sequencing data.** The raw dataset containing pair-ended reads with corresponding quality scores were merged and trimmed using the following settings, -fastq_minovlen 100, -fastq_maxee 2.0, -fastq_truncal 4, -fastq_minlen 130. De-replicating, purging from chimeric reads and constructing de novo zero-radius Operational Taxonomic Units (zOTU) was conducted using the UNOISE pipeline[82] coupled to the EZtaxon 16S rRNA gene collection as a reference database[83]. For downstream analyses the dataset (based on zOTUs phylotypes) was subsampled with 50,000 reads. Differences between experimental groups were evaluated using analysis of variance on distance matrices (Adonis). Differences in relative distribution among phylotypes were determined through Student's t-test, while correlations of phylotypes with measurements for insulin-stimulated glucose uptake were determined with Pearson Correlation Coefficients. These analyses were bootstrapped with 100 permutations and p-values corrected for Type I error with false discovery rate (FDR).

**Graphics.** The graphics shown in Figs. 1a and 2a, Supplementary Fig. 1a and the graphical abstract were made in ©BioRender - biorender.com (Toronto, Canada).

**Statistical analyses.** The data are expressed as mean ± SEM and individual data points (when applicable) and analyzed using GraphPad Prism 8. Statistical tests were performed using paired/non-paired t-tests or repeated/no-repeated two-way ANOVA as applicable. Multiple repeated Two-way ANOVAs were performed in analyses including all experimental groups testing for the effect of temperature within training groups or the effect of ET within each temperature. Sidak post-hoc test was performed when ANOVA revealed significant main effects and interactions. Microbiome analyses; For downstream analyses, the dataset (based on zOTUs phylotypes) was subsampled with 50,000 reads. Ordination Analyses were based on Bray-Curtis distances using Principal Coordinates Analysis (PCoA) and distance-based redundancy analysis (dbRDA), while differences between experimental groups were evaluated using analysis of similarities (ANOSIM test). Temporal effects were controlled using a Partial dbRDA (vegan 2.5-6 R-package), where the time of the experiment (Z-variable) was removed before analyzing the interactions between microbiome (X) and experimental groups (Y), e.g. dbRDA(X ~ Y+ Condition(Z)). Differences in relative distribution among phylotypes were determined through Student's t-test, while correlations of phylotypes with measurements for insulin-stimulated glucose uptake were determined with Pearson Correlation Coefficients. These analyses were bootstrapped with 100 permutations and p-values corrected for Type I error with FDR. The significance level was set at $\alpha = 0.05$.

**Reporting summary.** Further information on research design is available in the Nature Research Reporting Summary linked to this article.

## Data availability
The authors declare that all the data supporting the findings of this study are available within the paper, the supplementary figures or the source data provided. The source data underlying Figs. 1–6 and Supplementary Figs. 1–5 are provided as a Source Data file. Sequence data (in relation to Fig. 6 and Supplementary Fig. 5) are available at the European Nucleotide Archive, accession number ENA: PRJEB35066. Metadata and accession numbers are provided in the source data. Please contact the corresponding authors if questions in this regard.

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

## Acknowledgements

We acknowledge the skilled technical assistance of Betina Bolmgren and Irene Bech Nielsen (Molecular Physiology Group, Department of Nutrition, Exercise and Sports, University of Copenhagen, Denmark). We thank Professor Henriette Pilegaard (University of Copenhagen) and Professor D. Grahame Hardie (University of Dundee) for the kind donation of the PDH-1Eα antibody. L.S. and E.A.R. were supported by the Danish Council for Independent Research, Medical Sciences (grant DFF-4004-00233 to L.S., grant 6108-00203 to E.A.R.); The Novo Nordisk Foundation (grant 10429 to E.A.R., grant NNF16OC0023418 and NNF18OC0032082 to L.S.). C.H.O. is supported by the Danish Diabetes Academy, which is funded by the Novo Nordisk Foundation (grant number NNF17SA0031406). L.L.V.M. was supported by the PhD fellowship from The Lundbeck Foundation (grant 2015-3388 to L.L.V.M.).

## Author contributions

S.H.R., E.A.R., and L.S. conceptualized and designed the study. S.H.R. and L.S. conducted the experiments, performed the laboratory analysis, analyzed the data. S.H.R., E.A.R., and L.S. wrote the manuscript. C.H.O., I.K., M.A., L.L.V.M., W.K., J.L.C.M., D.S.N., and Z.G.H. all took part in conducting the experiments, performing laboratory analysis and/ or interpreting the data. All authors commented on and approved the final version of the manuscript. L.S. is the guarantor of this work and, as such, has full access to all the data in the study and takes responsibility for the integrity of the data and the accuracy of the data analyses.

## Competing interests

The authors declare no competing interests.
