## [Peer Review File · Nature Communications]

Reviewers' comments:

Reviewer #1 (Remarks to the Author):

A major claim of the manuscript is that the comprehensive usage of two temperatures (22°C or 30°C) comparisons provided novel findings, which were that thermoneutral housing of 30°C for mice blunted exercise-induced improvements in insulin action in muscle and adipose tissue; reduced the effects of training on energy expenditure, body composition, muscle and adipose tissue protein expressions, and the gut microbiome. The presented interpretations were that organismal adaptations to voluntary wheel-running training in mice critically depended upon housing temperature when voluntary running in wheels. The conclusions should be of interest to the general scientific field as parts of many mouse exercise studies are performed at room temperature, and thus could have portions of their data likely NOT translatable to humans!!!

The authors should consider amplifying the consequences of their sentence on lines 26-27 in their manuscript's Introduction. Lines 26-27 are: "They display sevenfold higher metabolic rate compared to humans...". The authors need to consider, so what does this imply about translatability? My interpretation of the author's lines 26-27 follow. The body to surface area of small animals increases as size decreases. To maintain thermoneutral body temperatures at 37°C, smaller sized animals have to have higher resting metabolic rates. As the authors' quotation, above, is that resting metabolic rates of mice are 7-fold higher than for humans. My examination of literature values found that the resting metabolic rate of mice approximates somewhere in the maximal oxygen uptake range of 60 ml/kg in the human. Furthermore, literature values for maximal O₂ uptake of mice on motor-drive treadmills are somewhere between 150-200 ml/kg/min. The assumption made by others is that the molecular events regulating a mouse's VO₂max at 150-200 ml/kg are exactly the same as the molecular events regulating humans at a VO₂max at 60-70 ml/kg. However, humans with VO₂max of 60-70 ml/kg compose only a small percentage of human population. VO₂max of most human adults range 20-30 ml/kg 30 yrs of age, and older. Please consider the above information in revising the impact of lines 26-27 in your important review under consideration.

The manuscript should influence the field. I expect many citations. The manuscript should influence regulators on exercise and non-exercise studies who still maintain temperatures of mouse rooms should be human values of 20-21° C. However, the authors should be careful as experiments in which mice are run to physical exhaustion likely should not be performed at 30°C in order not to produce heat exhaustion for mouse body temperatures greater than 40°C that would likely occur in forced treadmill running to physical exhaustion. Note, mice can be forced to run on treadmills for hours at 20°C. The end point for forced running to exhaustion by mice should be caused by energy metabolism, not by thermo-exhaustion, if the aim of the study is energy metabolism.

Reviewer #2 (Remarks to the Author):

I have seen this paper earlier, in a slightly different form. I accepted to review it again, as I already at the previous encounter felt that it was an important and interesting study, and I principally would like to promote its publication, as it is of potential importance for the interpretation of many studies on exercise effects examined in mice models.

Perhaps as a reviewer, one hopes that the efforts made will be reflected in manuscript improvement, even if the paper is rejected and submitted to another journal. This does not seem to have happened here. There are some rearrangements of the paper but I seem to recognize most of the points I earlier remarked upon as being unchanged in the present version. I could now go through in detail and identify each point. It is e.g. clear that the comment I made on what was

then fig. 2CD – the fat and lean content – now are relevant for fig. 1DE. And the authors have persisted in showing these data in this – in my opinion – somewhat confounding way. We could continue as this – but as the authors undoubtedly still also have the version I earlier reviewed, they could well respond to my earlier points although the numbering etc. has changed.

Thus, what I place below here is my earlier comments, and given that the authors take these points into consideration, I am of the opinion that this study is of a general significance that qualifies it for publication in Nature Communications.

This is principally an interesting study, examining the effects of thermoneutrality versus normal ambient temperature on the effects of exercise training in mice. While this may at first seem somewhat secondary, it may in reality be of large interest, as it for the first time examines the metabolic effects of exercise under what approaches human conditions. Increasingly it has been realized for many physiological (including metabolic) parameters that using thermoneutral conditions may profoundly alter the outcome of experiments, and these “new” outcomes may actually be those that are more relevant for translational studies. Similar studies have not earlier been performed for exercise. I therefore in general find this of interest for [omitted].

As one effect of thermoneutrality the authors find that the mice exercise less. Evidently this could be the explanation for the differential effects but the authors elegantly overcome this problem by pair-running and thus can dissect the effects of exercise volume from that of thermoneutrality as such. With some improvements, I would find the data of broad interest in the metabolic field – and the data are of course controversial in the entire discussion concerning health effects of exercise – an issue seemingly played down by the authors.

A general comment: the authors have some problems in distancing themselves from their data and obtain a broader view of the outcome. This is most “graphically” illustrated in the summary figures – the graphical abstract and fig 7, both of which are very much “ungraphical”. Here is much space for improvement: what do the authors really think are the main messages of their interesting investigations? And reading through the paper, the text becomes quite easily quite tedious. As far as I can see, the statements are correct one by one, but not every single effect on every single enzyme has to be given. The authors should try to write broader. For instance, nearly all % changes could be moved to the figure legends so that the text become broader. Also the bullet points are difficult to read.

Generally concerning the figures: the authors often use a truncated scale to show their results. Principally I am generally not fond of this as it disturbs the spontaneous evaluation of the results; additionally, for many of the data here the truncation has only a minor effect but its existence disturbs the view. Thus, the truncation should be removed from the following figures (I mean, the y-axes should start at 0 and be unbroken): !AEF, 2B(for CD see below)GHIJ, 3DGH, 4ABC,

Line 29: - this 7-fold by Schmidt-Nielsen probably really refers to the scaling effect and not to the issue of ambient temperature. Concerning the ambient temperature, the point must be that they are hypermetabolic as seen also from the data of the authors (and the quoted literature) by a factor of close to 2.

L 32: I don't think the mice are “over-eating” as they need the extra food to keep alive; they e.g. “need to eat and metabolize double as much food”

L 40: the argument comes very sudden here; try to reformulate.

L 49 and ahead: abbreviations are always to be avoided, although we accept some. I find ET and UT disturbing (except in figures); why not simply write “trained” (alt. exercised) and untrained all through.

L 81: 3-fold is not a slight increase – but the total daily activity is very low compared to the nightly activity – and this should be mentioned.

L 84: the figure 1N is not easy to understand and must be better explained.

L 91 and fig 2CD: I think that presenting lean and fat mass as % is misleading: when one goes up the other necessarily goes down. The actual masses as found in the supplement fig are the correct values and should replace these. I am doubtful that the authors should show the % values even in the suppl, as this type of representation is misleading.

L 95: not evident why just core temp was “importantly”

L 103: benefits of voluntary

L 107 and further: one way to see the glucose data at thermoneutrality is not to say that there was no effect of training – that sounds as a problem with thermoneutrality – but rather, as is the case according to the graphs, the GTT and the basal glucose values were already “better” in the untrained mice; training at 22 °C only improves these values to the levels already observed at thermoneutrality. The formulations throughout the paper should be reformulated in this respect.

L 156: retro-orbital

L 159 and ahead: I am not fully happy with the glucose uptake experiments in that only accumulation of labelled glucose was followed – there was no marker for trapped plasma in the tissues. Thus, values may be secondary to blood flow etc. In which way have the authors controlled for this? And the uptakes are referred to as insulin-stimulated uptakes. How do the authors know? All experiments were performed in the presence of insulin so the uptakes may not necessarily be insulin-stimulated at all – and the effects of training could be on non-insulin-mediated uptake?

L 184: should the surprise not only be that it is there even in the pair-exercised mice?

L 207 and ahead: as the authors have both uptake per mg and total weights of the different tissues should they not calculate (or present) total tissue uptakes?

Fig 4H title: mitochondrial

L 227 and ahead: I am very surprised that the authors have not included UCP1 among the proteins they have examined in the different adipose tissues. They have the protein samples and they should be able to obtain UCP1 antibodies. I think this is an important addition to the paper, considering the large interest in the effect of exercise on BAT and beige fat – and particularly in relation to the irisin effects that the authors themselves refer to here. Thus, these data should be added to the main figures.

Figs 4 and 5: I am not fully happy with the presentation of the mitochondrial (and other) proteins given relative to 22 untrained. The change in scales on the different panels gives very confusing impressions (cf. e.g. fig 5 I v J complex I. Also we get no feeling for the relative amounts between the different tissues. At least one could keep same scale for most (up to 4) except for I complex I.

L 249 and ahead: also GM is an abbreviation without purpose.

L 272 and ahead: often discussion sections improve by having subheadings with clear statements

L 285 and ahead: again, the presentation of the training effects on glucose/insulin metabolism is

discussable in that the thermoneutrality mice are already fine. Thus, the conclusion must more or less be that the beneficial effects of training on these parameters is an experimental artefact caused by the "disturbed" metabolism caused by the increased metabolism in the cold.

L 333: observed

Also in the discussion I think the authors should try to distance themselves somewhat from the specific points and make it more clear what the perspectives are. For me, the observation that the wheel running interest is practically lost at thermoneutrality is very interesting as this behavior has always been put forward as very remarkable. Do they really do it because they freeze? And, of course, humans tend to exercise with less clothes on, i.e. below thermoneutrality. Is this important? Is the present paper really a heavy argument against the (mouse experimentally described) benefits of exercise on metabolic health? Etc.

And the authors should reconsider the abstract in the light of all points above.

Reviewer #3 (Remarks to the Author):

This study aimed to assess effects of exercise training (ET) on metabolic parameters at thermoneutral (30°C) or regular (22°C) conditions. The authors observed about 60% decrease in voluntary ET amount at 30°C compared to 22°C and the effect of ET on glucose tolerance was absent at 30°C. The authors then correctly evaluated effects of ET at 20°C that matched the amount of the 30°C. In the latter case, there was no effect of ET at 22°C as in the group at 30°C. Therefore, from this point on, the study should have compared between those 2 groups and not with the 22°C ET mice with unlimited ET, i.e., comparison of the groups at 30°C without/with ET, and at 22°C without/with paired ET volume would be the appropriate approach. Otherwise, all the comparisons to the 22°C with non-matching ET is not interpretable because there are two factors that are varied at the same time (ToC and ET). This includes several results including the microbiota analysis and the molecular mechanisms where the paired ET group is missing.

A few other comments:

1. This study addresses how housing temperature influences adaptations following exercise training. There are 2 factors here, exercise and temperature. Without showing the p values for the interaction effects of these 2 factors, it's hard to claim that the effects of exercise are temperature dependent.
2. The study needs to be better organized. The authors should focus on how mice response to exercise training in different housing temperature, not the effects of temperature or running volume, which could moved to a separate section or supplementary results.
3. Mice at 30°C eat less than mice at 22°C, which might explain the lower levels of blood glucose. Not necessarily the direct effect of housing temperature (line 71-73).
4. A previous study that did not report significant decrease in mouse activity at higher ToC and comprehensively characterized several relevant parameters and factors was not cited (Abreu-Vieira, Mol Metab. 2015 Jun; 4(6): 461-470).

Response to reviewers. We thank the reviewers for their thorough and constructive comments to our manuscript. We have taken all of the comments into consideration and have made substantial changes to the manuscript as well as adding novel data. We have responded to each reviewer below.

Reviewer #1 (Remarks to the Author):

A major claim of the manuscript is that the comprehensive usage of two temperatures (22°C or 30°C) comparisons provided novel findings, which were that thermoneutral housing of 30°C for mice blunted exercise-induced improvements in insulin action in muscle and adipose tissue; reduced the effects of training on energy expenditure, body composition, muscle and adipose tissue protein expressions, and the gut microbiome. The presented interpretations were that organismal adaptations to voluntary wheel-running training in mice critically depended upon housing temperature when voluntary running in wheels. The conclusions should be of interest to the general scientific field as parts of many mouse exercise studies are performed at room temperature, and thus could have portions of their data likely NOT translatable to humans!!!

We thank the reviewer for above comments of your manuscript and we agree that our results significantly influence the way we interpret current and future data obtained using this model, as well as the results translatability to the human condition.

The authors should consider amplifying the consequences of their sentence on lines 26-27 in their manuscript's Introduction. Lines 26-27 are: "They display sevenfold higher metabolic rate compared to humans...". The authors need to consider, so what does this imply about translatability? My interpretation of the author's lines 26-27 follow. The body to surface area of small animals increases as size decreases. To maintain thermoneutral body temperatures at 37°C, smaller sized animals have to have higher resting metabolic rates. As the authors' quotation, above, is that resting metabolic rates of mice are 7-fold higher than for humans. My examination of literature values found that the resting metabolic rate of mice approximates somewhere in the maximal oxygen uptake range of 60 ml/kg in the human. Furthermore, literature values for maximal O₂ uptake of mice on motor-drive treadmills are somewhere between 150-200 ml/kg/min. The assumption made by others is that the molecular events regulating a mouse's VO₂max at 150-200 ml/kg are exactly the same as the molecular events regulating humans at a VO₂max at 60-70 ml/kg. However, humans with VO₂max of 60-70 ml/kg compose only a small percentage of human population. VO₂max of most human adults range 20-30 ml/kg 30 yrs of age, and older. Please consider the above information in revising the impact of lines 26-27 in your important review under consideration.

These are very good observations! In accordance with reviewer #2, this sentence has been altered, as the absolute differences are approximately 2-fold and not 7-fold as originally written (line; 27). This is also approximately the effect size we observe in our own study (Figure 1C).

About the implication of the total maximal oxygen uptake, and the obvious difference between exercise in mice and in humans, we have now included this in the discussion as well (line; 282-291).

The manuscript should influence the field. I expect many citations. The manuscript should influence regulators on exercise and non-exercise studies who still maintain temperatures of mouse rooms should be

human values of 20-21° C. However, the authors should be careful as experiments in which mice are run to physical exhaustion likely should not be performed at 30°C in order not to produce heat exhaustion for mouse body temperatures greater than 40°C that would likely occur in forced treadmill running to physical exhaustion. Note, mice can be forced to run on treadmills for hours at 20°C. The end point for forced running to exhaustion by mice should be caused by energy metabolism, not by thermo-exhaustion, if the aim of the study is energy metabolism.

The maximal running test in our study was not performed at 30°C but at 22°C for all mice. We chose this to be able to randomize the experimental groups, as well as being able to perform this test blinded (one person placing the mice on the treadmills, another person (blinded) to determine the exhaustion of the individual mouse). This has been clarified in the methods description (line; 389-395). As such, heat exhaustion was likely not a confounding factor in our study. However, in relation to your comment, McKie et al., in a recent study in The Journal of Physiology ran male C57bl/6 mice to exhaustion in 29°C <https://doi.org/10.1113/JP278221>. The authors did not see a difference in running time to exhaustion (95-115 minutes) and that study has now been included in this very relevant discussion (line; 300-302).

We thank the reviewer for the excellent comments and we believe that the more in depth discussion and elaboration of the use of our model, has significantly improved the manuscript.

Reviewer #2 (Remarks to the Author):

I have seen this paper earlier, in a slightly different form. I accepted to review it again, as I already at the previous encounter felt that it was an important and interesting study, and I principally would like to promote its publication, as it is of potential importance for the interpretation of many studies on exercise effects examined in mice models.

Perhaps as a reviewer, one hopes that the efforts made will be reflected in manuscript improvement, even if the paper is rejected and submitted to another journal. This does not seem to have happened here. There are some rearrangements of the paper but I seem to recognize most of the points I earlier remarked upon as being unchanged in the present version. I could now go through in detail and identify each point. It is e.g. clear that the comment I made on what was then fig. 2CD – the fat and lean content – now are relevant for fig. 1DE. And the authors have persisted in showing these data in this – in my opinion – somewhat confounding way. We could continue as this – but as the authors undoubtedly still also have the version I earlier reviewed, they could well respond to my earlier points although the numbering etc. has changed.

Thus, what I place below here is my earlier comments, and given that the authors take these points into consideration, I am of the opinion that this study is of a general significance that qualifies it for publication in Nature Communications.

We thank the reviewer, for the positive feedback. We appreciate that you can see this study as having a general impact of the exercise field. We thank the reviewer for the previous highly in-depth review of the manuscript, from which we did indeed take all of the comments into consideration and have revised accordingly before submitting to Nature Communication. We have now further extensively revised the manuscript, including changing the MRI-data e.g. and have completely restructured the graphs and results section.

Thank you.

This is principally an interesting study, examining the effects of thermoneutrality versus normal ambient temperature on the effects of exercise training in mice. While this may at first seem somewhat secondary, it may in reality be of large interest, as it for the first time examine the metabolic effects of exercise under what approaches human conditions. Increasingly it has been realized for many physiological (including metabolic) parameters that using thermoneutral conditions may profoundly alter the outcome of experiments, and these “new” outcomes may actually be those that are more relevant for translational studies. Similar studies have not earlier been performed for exercise. I therefore in general find this of interest for [omitted].

As one effect of thermoneutrality the authors find that the mice exercise less. Evidently this could be the explanation for the differential effects but the authors elegantly overcome this problem by pair-running and thus can dissect the effects of exercise volume from that of thermoneutrality as such. With some improvements, I would find the data of broad interest in the metabolic field – and the data are of course controversial in the entire discussion concerning health effects of exercise – an issue seemingly played down by the authors.

Thank you for this comment and we agree that our data are seemingly controversial. We did not mean to play this down, however we also felt, that we could not directly compare this model to the human situation. However, we have now extensively extended and revised the discussion. We have included additional references showing that this model might in fact mimic lean healthy humans where an increased insulin sensitivity or improved glucose tolerance following exercise training is often not observed (line; 269-271).

We have done this based on this reviewer's suggestions and hope that it has helped to better extrapolate our findings to humans. We believe that it has strengthened the discussion on the implications for translationability of our human model to humans. We do believe our thermoneutrally-housed training model better mimics the human condition and this argument has been strengthened in the discussion.

A general comment: the authors have some problems in distancing themselves from their data and obtain a broader view of the outcome. This is most "graphically" illustrated in the summary figures – the graphical abstract and fig 7, both of which are very much "ungraphical". Here is much space for improvement: what do the authors really think are the main messages of their interesting investigations? And reading through the paper, the text becomes quite easily quite tedious. As far as I can see, the statements are correct one by one, but not every single effect on every single enzyme has to be given. The authors should try to write broader. For instance, nearly all % changes could be moved to the figure legends so that the text become broader.

Also the bullet points are difficult to read.

We have addressed this by completely reformatting the results description, graphs presentations, the bullet points, and restructured the graphical abstract (obs. Fig. 7 was already removed in the manuscript). We believe this has helped obtain a broader view of the outcome of the study, and we thank the reviewer for providing us with this feedback.

Generally concerning the figures: the authors often use a truncated scale to show their results. Principally I am generally not fond of this as it disturbs the spontaneous evaluation of the results; additionally, for many of the data here the truncation has only a minor effect but its existence disturbs the view. Thus, the truncation should be removed from the following figures (I mean, the y-axes should start at 0 and be unbroken): !AEF, 2B(for CD see below)GHII, 3DGH, 4ABC,

All truncated y-axes have now been changed to full lines.

Line 29: - this 7-fold by Schmidt-Nielsen probably really refers to the scaling effect and not to the issue of ambient temperature. Concerning the ambient temperature, the point must be that they are hypermetabolic as seen also from the data of the authors (and the quoted literature) by a factor of close to 2.

Thank you for spotting this, we agree with the reviewer and we have changed the text accordingly (line; 27).

L 32: I don't think the mice are "over-eating" as they need the extra food to keep alive; they e.g. "need to eat and metabolize double as much food"

This sentence had already been removed from the first version of the manuscript, as we also mostly believe that the increase food intake is in order to sustain the core temperature and health at ambient

temperatures.

L 40: the argument comes very sudden here; try to reformulate.

This paragraph has been completely reformulated, and this line has also been changed (line; 25-27)

L 49 and ahead: abbreviations are always to be avoided, although we accept some. I find ET and UT disturbing (except in figures); why not simply write “trained” (alt. exercised) and untrained all through.

We chose to use UT and ET in the main text, as the text would become too heavy with too many repetitive words. Furthermore, we want for the text description to align with that in the figures (in the latter where there is no room to write “untrained” or trained” below each bar). However, based on this reviewers comment, we have reduced the usage of UT and ET from the main text in the new manuscript.

L 81: 3-fold is not a slight increase – but the total daily activity is very low compared to the nightly activity – and this should be mentioned.

This sentence has been changed accordingly. In addition, we have included a discussion regarding habitual activity and the changes with housing temperature (line; 71-75).

L 84: the figure 1N is not easy to understand and must be better explained.

In order to increase the understanding of the manuscript and results, this graph has been removed from the previous submission.

L 91 and fig 2CD: I think that presenting lean and fat mass as % is misleading: when one goes up the other necessarily goes down. The actual masses as found in the supplement fig are the correct values and should replace these. I am doubtful that the authors should show the % values even in the suppl, as this type of representation is misleading.

This has been changed. We now show the raw values in gram, and the relative changes in body composition (body fat) was also calculated from these data. Please see Figure 2E-G.

L 95: not evident why just core temp was “importantly”

This sentence has been changed (line; 77-83).

L 103: benefits of voluntary

This has been corrected.

L 107 and further: one way to see the glucose data at thermoneutrality is not to say that there was no effect of training – that sounds as a problem with thermoneutrality – but rather, as is the case according to the graphs, the GTT and the basal glucose values were already “better” in the untrained mice; training at 22 °C only improves these values to the levels already observed at thermoneutrality. The formulations throughout the paper should be reformulated in this respect.

We agree with the reviewer on this matter – we have now emphasized in the discussion and description of the results that the thermoneutrality-housed mice are indeed more glucose tolerant than ambient-housed mice. (line; 111-117 & 278-281)

L 156: retro-orbital

This has been corrected.

L 159 and ahead: I am not fully happy with the glucose uptake experiments in that only accumulation of labelled glucose was followed – there was no marker for trapped plasma in the tissues. Thus, values may be secondary to blood flow etc. In which way have the authors controlled for this? And the uptakes are referred to as insulin-stimulated uptakes. How do the authors know? All experiments were performed in the presence of insulin so the uptakes may not necessarily be insulin-stimulated at all – and the effects of training could be on non-insulin-mediated uptake?

Our 2-DG method does specifically measure tissue-specific glucose uptake, as only glucose taken up is metabolized and converted into glucose-6 phosphate (G-6-P). G-6-P is trapped inside cell, thus our measurement of ³H-2DG-G-6-P, reflects only the amount of glucose that has actually entered the cell. However, as this is a rather complicated analysis that we have now further elaborated on in the methods section. (line; 425-432)

That the uptakes referred to are indeed insulin-stimulated, we know because we did also analyze basal 2-DG uptake in all tissue reported. As these results revealed no differences between interventions, we are confident that the differences observed, do in fact relate to insulin-stimulated glucose uptake. Basal measurements are included in supplementary data (Suppl. Fig.3C). All plasma tracer appearances are reported in the supplementary data (Suppl. Fig.3D) as well, which importantly did not differ between groups.

We hope our additions and methodological clarifications have addressed these concerns.

L 184: should the surprise not only be that it is there even in the pair-exercised mice?

We are unsure as to what the reviewer is referring to here. If it is, that it is surprising there is an increase in the expression of complexes of the electron transport chain, then that is not a surprise, that is shown by numerous studies.

L 207 and ahead: as the authors have both uptake per mg and total weights of the different tissues should they not calculate (or present) total tissue uptakes?

This is an interesting suggestion. However, the manuscript is already extremely busy with figures and we do not see that this analysis would add extra information to the current study, nor would it potentially influence any of the conclusions drawn. Thus, we respectfully have decided not to include these calculations. Below are the calculated values.

Fig 4H title: mitochondrial

This has now been corrected.

L 227 and ahead: I am very surprised that the authors have not included UCP1 among the proteins they have examined in the different adipose tissues. They have the protein samples and they should be able to obtain UCP1 antibodies. I think this is an important addition to the paper, considering the large interest in the effect of exercise on BAT and beige fat – and particularly in relation to the irisin effects that the authors themselves refer to here. Thus, these data should be added to the main figures.

This is an excellent suggestion and we have now analyzed UCP1 in WAT and BAT and included those measurements in the main figure (Fig. 5C and 5D). UCP1 did not change in response to exercise training in brown adipose tissue but was generally markedly downregulated in response to thermoneutral housing, as expected. In WAT, training increased UCP1 protein in both housing temperatures, but surprisingly not in the paired 22C mice. These data are in good agreement with our results from qPCR analyses (Suppl. 5A and 5B).

Figs 4 and 5: I am not fully happy with the presentation of the mitochondrial (and other) proteins given relative to 22 untrained. The change in scales on the different panels gives very confusing impressions (cf. e.g. fig 5 I v J complex I. Also we get no feeling for the relative amounts between the different tissues. At least one could keep same scale for most (up to 4) except for I complex I.

We have now changed all of the graphs based on this reviewer's suggestion and we hope this has made the data presentation more clear.

L 249 and ahead: also GM is an abbreviation without purpose.

This has been changed and we do not abbreviate "gut microbiome".

L 272 and ahead: often discussion sections improve by having subheadings with clear statements

Due to the guidelines of Nature Communication, we are not allowed to include subheadings in the discussion: "[...] the Discussion should be succinct and may not contain subheadings."

L 285 and ahead: again, the presentation of the training effects on glucose/insulin metabolism is discussable in that the thermoneutrality mice are already fine. Thus, the conclusion must more or less be

that the beneficial effects of training on these parameters is an experimental artefact caused by the “disturbed” metabolism caused by the increased metabolism in the cold.

This is indeed a good point and we have now elaborated more on this in the discussion and stated this possibility when investigating lean mice. The picture could be different when studying metabolically challenged mice, such as DIO or ob/ob mice and this possibility has also now been discussed (line; 282-291).

L 333: observed

This has been corrected.

Also in the discussion I think the authors should try to distance themselves somewhat from the specific points and make it more clear what the perspectives are. For me, the observation that the wheel running interest is practically lost at thermoneutrality is very interesting as this behavior has always been put forward as very remarkable. Do they really do it because they freeze? And, of course, humans tend to exercise with less clothes on, i.e. below thermoneutrality. Is this important? Is the present paper really a heavy argument against the (mouse experimentally described) benefits of exercise on metabolic health? Etc.

We have now included a much more comprehensive discussion on this model's use in mimicking the human condition. This issue was also raised by one of the other reviewers and we have done extensive revisions to the discussion. We believe that those changes have provided more perspective to our findings and their impact.

And the authors should reconsider the abstract in the light of all points above.

The abstract has been modified accordingly, in addition to the changes to the previous submitted manuscript. (See highlight on front page 1)

We wish to thank the reviewer for insightful comments and suggestions. Our extra analyses, deeper methods descriptions, as well as the much more comprehensive discussion, based on this reviewers comments, has markedly improved the quality of the manuscript.

Reviewer #3 (Remarks to the Author):

This study aimed to assess effects of exercise training (ET) on metabolic parameters at thermoneutral (30oC) or regular (22oC) conditions. The authors observed about 60% decrease in voluntary ET amount at 30oC compared to 22oC and the effect of ET on glucose tolerance was absent at 30oC. The authors then correctly evaluated effects of ET at 20oC that matched the amount of the 30oC. In the latter case, there was no effect of ET at 22oC as in the group at 30oC.

Therefore, from this point on, the study should have compared between those 2 groups and not with the 22oC ET mice with unlimited ET, i.e., comparison of the groups at 30oC without/with ET, and at 22oC without/with paired ET volume would be the appropriate approach. Otherwise, all the comparisons to the 22oC with non-matching ET is not interpretable because there are two factors that are varied at the same time (ToC and ET). This includes several results including the microbiota analysis and the molecular mechanisms where the paired ET group is missing.

This is a good suggestion and we have now included the matching ET in all of the analyses and main graphs for direct comparison and easier data interpretation. Also, we have re-sequenced the microbiota and included the matching ET in the analysis. Consequently, the entire manuscript has been restructured, which we believe has markedly improved the presentation of the results and clarity.

A few other comments:

1. This study addresses how housing temperature influences adaptations following exercise training. There are 2 factors here, exercise and temperature. Without showing the p values for the interaction effects of these 2 factors, it's hard to claim that the effects of exercise are temperature dependent.

We have modified the data presentations to better reflect the statistically significant effects of training and temperature. Considering the other reviewers' comments, we did not include all the p-values, as this would make the manuscript heavy to read and interpret. This was a direct concern of reviewer #2.

2. The study needs to be better organized. The authors should focus on how mice response to exercise training in different housing temperature, not the effects of temperature or running volume, which could moved to a separate section or supplementary results.

In light of the comments also provided by reviewer 2, we have reorganized the study to better focus on the exercise training responses at different housing temperatures. This is done by including the ½ ET data in all of the graphs (excluding figure 1) and clearly stating the effects in the main text.

3. Mice at 30oC eat less than mice at 22C, which might explain the lower levels of blood glucose. Not necessarily the direct effect of housing temperature (line 71-73).

This is a possible explanation that has now been included in the discussion (line; 341-343). Thank you for this observation.

4. A previous study that did not report significant decrease in mouse activity at higher ToC and comprehensively characterized several relevant parameters and factors was not cited (Abreu-Vieira, Mol Metab. 2015 Jun; 4(6): 461–470).

We have indeed read this paper. The reason we initially did not see the necessity to cite that study, was due to critical differences in the housing method. Most importantly, Abreu-Vieira et al. (2015) do rapid/acute changes to the housing temperature from day-to-day. The following sentence is from their paper; “Mice were acclimated to the chambers for 3 days at 22°C, followed in order by one day each at 22°C, 26°C, 30°C, 33°C, 28°C, 24°C, 18°C, 12°C, and 4°C...”.

The fact that they do not see any differences in activity between 22°C and 30°C during acute changes in housing temperature fits nicely with our data presented in supplementary figure 1E. In this experiment (Suppl. Fig 1B-E), we housed mice in 22°C in metabolic chambers before the temperature was raised to 30°C. While food intake and oxygen consumption decreased along with the increased housing temperature, the habitual activity was not acutely affected. These data show that longer acclimatization time to different housing temperatures is important depending on the investigated parameters. Despite those differences, we do see the relevance and have now added the paper by Abreu-vieira et al (2015), when discussing our own data on the acute change in housing temperature (line; 333-336).

We hope that the restructuring of the graphs as well as the more comprehensive discussion on how this model relates to the human condition has improved the clarity and focus of our investigation. We thank the reviewer for excellent feedback.

Reviewers' comments:

Reviewer #1 (Remarks to the Author):

Thank you for your alterations.

Reviewer #2 (Remarks to the Author):

The authors have now substantially improved the paper. I can principally approve of publication provided that the authors consider the following points.

Most importantly: Many formulations throughout promote the feeling that the authors want to show that studying mice at 30 is not beneficial because certain training effects disappear. E.g. the subheading line 194 is "Adipose tissue molecular adaptations to exercise training are partially reduced by thermoneutral housing". Is not a more physiological adequate formulation: "Reported effects of exercise training on a.t.m.a. are mainly due to maintaining mice at cold-stressed conditions" – or similar.

Similarly, the first subheading: "Thermoneutral housing lowers energy expenditure and metabolic fluctuations.....". How about: Ambient temperature increases energy expenditure and metabolic fluctuations..."

The general impression of the formulations is thus that it is "bad" to do experiments at 30 because the exercise effects then disappear – but is the conclusion not rather that to do exp at 22 is "bad" because the mice are cold-stressed? This is particular evident at the bullet points that all sounds as "30 is not good". Should not the last bullet point be something like "Thus, many reported effects of exercise are likely secondary to the metabolic stress that mice experience at 22 °C, making it doubtful to translate the effects to human conditions" – or similar. When the authors reread their text, they should think about this and decide what message they want to give: 30 is not good or 30 is better? (or at least: it matters what temperature is used and conclusions can be qualitatively different).

A second issue is related to the glucose uptake data.

Firstly, concerning the methods: despite what the authors state, it is not well described. In the methods, 2 references are given to the method, but these 2 references are not in the reference list (!). I found them anyhow, I believe, and I find that the following sentences clearly are missing from the description, to make it acceptable that the authors really measure glucose uptake and not just glucose associated with the tissue: "Muscle samples were weighed and homogenized in 0.5% perchloric acid. Homogenates were centrifuged and neutralized with KOH. One aliquot was counted directly to determine [2-3H]DG and [2-3H]DGP radioactivity. A second aliquot was treated with Ba(OH)₂ and ZnSO₄ to remove [2-3H]DGP and any tracer incorporated into the glycogen and then counted to determine [2-3H]DG radioactivity. [2-3H]DGP is the difference between the two aliquots."

Secondly, I am now doubtful whether the data presented as insulin-stimulated glucose uptake (fig. 3 D-H) really is this. Is it not total uptake? I.e. should not the data in suppl fig 3C be subtracted from the data in fig. 3 to make it truly "insulin-stimulated"? At least this must be clarified. Does it influence the conclusions?

Thirdly, I am not happy with the conclusion of the authors not to show the total uptakes (now only provided as an extra graph to me). The point is that it would from that – and the underlying data in fig 3 – glucose uptake and tissue weight – be doubtful what is concluded from fig 3 concerning effects of ET on glucose uptake. As far as I can see, the effect of ET is solely on tissue composition – less fat – so that there is an apparent effect on glucose uptake because the fat dilutes the water

space (cytoplasm per mg tissue). In reality, as calculated by the authors, there is no effect of ET on glucose uptake into iWAT, eWAT, BAT and probably pWAT. This I think is the correct conclusion concerning these tissues. Unfortunately, the authors apparently did not collect total tissue weight for muscle, so the same analysis cannot be made for the muscle tissues. Further, page 8, I think that it is most natural to first discuss the differences in initial blood glucose levels and then the actual GTT.

Finally, the same reference appears as ref. 26 and ref. 59.

Reviewer #3 (Remarks to the Author):

I appreciate the authors including several comparisons with paired ET group and sequencing microbiota of all mice.

However, one concern regarding this is that microbiota of two UT groups at 22oC (dark blue and green) does not cluster together in Fig 6A, although they are technically the same. Rather blue are closer to red (30oC) and both of those are apart from all green. Is there a technical issue that can explain this? Otherwise, the statement that housing Temp is the main factor in driving changes in microbiome is not supported.

For Figs. 6B-E, indication of which exact fold changes are shown is needed. This can be done by labels under x axis. For example, fig 6B, is it 30oC UT/22oC UT? but then, is it 22oC dark blue or green?

line 230: it's not diversity what is shown.

Rebuttal to reviewer comments:

We thank the reviewers for an additional round of constructive criticism. Below we provide answers to each individual comment.

Reviewers' comments:

Reviewer #1 (Remarks to the Author): Thank you for your alterations.

We are happy that our corrections to the manuscript were sufficient for the reviewer. Thank you.

Reviewer #2 (Remarks to the Author):

The authors have now substantially improved the paper. I can principally approve of publication provided that the authors consider the following points.

Thank you for undertaking this second round of review and for your constructive additional comments. We have addressed all of the additional points and believe that the manuscript has been strengthened by it.

Most importantly: Many formulations throughout promote the feeling that the authors want to show that studying mice at 30 is not beneficial because certain training effects disappear. E.g. the subheading line 194 is "Adipose tissue molecular adaptations to exercise training are partially reduced by thermoneutral housing". Is not a more physiological adequate formulation: "Reported effects of exercise training on a.t.m.a. are mainly due to maintaining mice at cold-stressed conditions" – or similar.

Similarly, the first subheading: "Thermoneutral housing lowers energy expenditure and metabolic fluctuations.....". How about: Ambient temperature increases energy expenditure and metabolic fluctuations..."

The general impression of the formulations is thus that it is "bad" to do experiments at 30 because the exercise effects then disappear – but is the conclusion not rather that to do exp at 22 is "bad" because the mice are cold-stressed? This is particular evident at the bullet points that all sounds as "30 is not good". Should not the last bullet point be something like "Thus, many reported effects of exercise are likely secondary to the metabolic stress that mice experience at 22 °C, making it doubtful to translate the effects to human conditions" – or similar. When the authors reread their text, they should think about this and decide what message they want to give: 30 is not good or 30 is better? (or at least: it matters what temperature is used and conclusions can be qualitatively different).

These are good points, as we do not mean to state that 30 C housing is not a good housing temperature for studying ET adaptations in mice. Conversely, we have argued in the discussion that the lack of adaptations in 30C housed mice in fact are more similar to the minor effects on insulin sensitivity and glucose tolerance observed in lean young healthy humans. This was included in the discussion based on the first round of reviewer comments. However, we can see that the formulations can still be improved and we have now modified accordingly by including it in a bullet, modified the abstract, added this point to the conclusion, as well as pointed this out throughout the main text.

A second issue is related to the glucose uptake data.

Firstly, concerning the methods: despite what the authors state, it is not well described. In the methods, 2 references are given to the method, but these 2 references are not in the reference list (!). I found them anyhow, I believe, and I find that the following sentences clearly are missing from the description, to make it acceptable that the authors really measure glucose uptake and not just glucose associated with the tissue: "Muscle samples were weighed and homogenized in 0.5% perchloric acid. Homogenates were centrifuged and neutralized with KOH. One aliquot was counted directly to determine [2-3H]DG and [2-3H]DGP radioactivity. A second aliquot was treated with Ba(OH)₂ and ZnSO₄ to remove [2-3H]DGP and any tracer incorporated into the glycogen and then counted to determine [2-3H]DG radioactivity. [2-3H]DGP is the difference between the two aliquots."

We have corrected the reference mistake, thank you for spotting that. To ensure clarity in the method section, we have now significantly elaborated on the methods section (line: 446-452). We are certain that we measure uptake, as we analyze the intramuscular [³H]2DG-phosphate, which is separated from the non-phosphorylated [³H]2DG, that is located in the blood and the extracellular space.

Secondly, I am now doubtful whether the data presented as insulin-stimulated glucose uptake (fig. 3 D-H) really is this. Is it not total uptake? I.e. should not the data in suppl fig 3C be subtracted from the data in fig. 3 to make it truly "insulin-stimulated"? At least this must be clarified. Does it influence the conclusions?

In this in vivo experimental setup, we unfortunately cannot measure both basal and insulin-stimulated glucose uptake in the same mouse, although that would have been optimal. Because of that, we cannot relate the insulin-stimulated glucose uptake to the baseline for each mouse, as we do not know the baseline for the individual mouse. Clearly, the reviewer is correct and insulin-stimulated glucose uptake reflects glucose uptake in the insulin-stimulated state, which also includes basal uptake. The same argument holds for intracellular signaling and any other insulin-stimulated measurement but given that we do not have the possibility of a within-mouse paired design, it will be incorrect to subtract baseline values (as is also the case in any other experiment performed in vivo using any stimuli). Accordingly, we are careful not to state in the text, that it is an uptake above the basal glucose uptake.

Thirdly, I am not happy with the conclusion of the authors not to show the total uptakes (now only provided as an extra graph to me). The point is that it would from that – and the underlying data in fig 3 – glucose uptake and tissue weight – be doubtful what is concluded from fig 3 concerning effects of ET on glucose uptake. As far as I can see, the effect of ET is solely on tissue composition – less fat – so that there is an apparent effect on glucose uptake because the fat dilutes the water space (cytoplasm per mg tissue). In reality, as calculated by the authors, there is no effect of ET on glucose uptake into iWAT, eWAT, BAT and probably pWAT. This I think is the correct conclusion concerning these tissues. Unfortunately, the authors apparently did not collect total tissue weight for muscle, so the same analysis cannot be made for the muscle tissues.

Just to clarify, we changed the abbreviation of eWAT to pWAT, as only male mice have epididymal fat pads, where the equivalent depot is called periovarian fat pads in female mice. To ensure no confusion (although we did not change the name in the graphs we send in the 1. revision), we now call this fat pad the perigonadal fat pad (pWAT) as it constitutes the same type fat pad in both male and female mice which were dissected (PMID: 27148535). Sorry for any misunderstanding.

The point is good. We have now included these analyses/calculations in the manuscript along with a description of what that means for the impact of WAT (line: 170-174, suppl. Fig. 3E).

We do believe that the insulin-stimulated glucose uptake should be presented in the main figure relative to the weight of tissue (Fig. 3). There are several reasons for this; Firstly, this is a commonly used way to report glucose uptake measured by tracer (references from different labs; PMID: 30400016, 25605808, 30472415), allowing for direct comparisons across articles and labs. Secondly, we do not believe that the fat tissue measurements are diluted. Regarding the reviewer's comment on dilution of fat tissue, then this increase in TG content would perhaps lead to a dilution of measurements of glucose uptake measured with tracers as in the current study. This is however not true, as basal glucose uptake in different adipose tissue from high-fat diet fed mice is not different from mice fed a regular chow diet (as done in current investigation). This has been shown by our self and other laboratories (PMID: 29749029 & 30400016). We therefore do not believe that the fat tissue measurements in current investigation are diluted.

Regarding the muscle-issue, it is unfortunate that we did not weigh all of the muscles; however, this was not feasible within the limitations of the experimental set-up, the experimental tracer design being time-sensitive. However, in other exercise training studies in our lab, both gastrocnemius (+25%) and tibialis anterior (+19%) muscle weight increase after voluntary wheel running in female C57bl6 mice (unpublished). Thus, the effect size of ET on insulin-stimulated glucose uptake would be even greater. We are sorry we cannot make this calculation, however this is due to methodological considerations of the experimental setup. The unpublished data can be seen below.

We hope that these corrections will satisfy the reviewer.

Further, page 8, I think that it is most natural to first discuss the differences in initial blood glucose levels and then the actual GTT.

Thank you for this suggestion, and we agree that it makes sense to initiate the description of the GTT results with the fasting blood glucose data (line: 110-121).

Finally, the same reference appears as ref. 26 and ref. 59.

Thank you, this has been corrected

Reviewer #3 (Remarks to the Author):

I appreciate the authors including several comparisons with paired ET group and sequencing microbiota of all mice.

However, one concern regarding this is that microbiota of two UT groups at 22oC (dark blue and green) does not cluster together in Fig 6A, although they are technically the same. Rather blue are closer to red (30oC) and both of those are apart from all green. Is there a technical issue that can explain this?

Otherwise, the statement that housing Temp is the main factor in driving changes in microbiome is not supported.

This is a good point and we acknowledge the reviewer's concern about GM clustering. On one hand we agree that those independent experiments units at 22C are technically the same, but on the other this is not necessarily the case for GM members. GM composition is highly dynamic and changes over time and batch effects are often expected and are well described in the literature [e.g. Randall et al, 2019 (PMID: 31477171), Gibbons et al., 2018 (PMID: 29684016)]. However, to avoid misinterpretations we have adjusted the effect of time-batch in our GM analyses and made changes accordingly throughout the manuscript, e.g. lines 239-243, Figure 6A and Supplementary figure 5C. The results show discrimination of experimental units and clustering of those units according to the housing temperatures. We strongly believe these changes improve significantly the comparability and context of the GM section. The results and the conclusions remain the same.

For Figs. 6B-E, indication of which exact fold changes are shown is needed. This can be done by labels under x axis. For example, fig 6B, is it 30oC UT/22oC UT? but then, is it 22oC dark blue or green?

This is a very good suggestion. To ensure clarity, we have added a label (color codes) to each of the volcano plots (see Fig. 6B-E).

line 230: it's not diversity what is shown.

Thank you for pointing out this mistake, and it is has now been corrected (line: 239)

REVIEWERS' COMMENTS:

Reviewer #3 (Remarks to the Author):

Thank you for your responses.

One last comment: please add to the methods section a more detailed description of how exactly the adjustment for time/batch effect in microbiota analysis was done in fig.6.

REVIEWERS' COMMENTS:

Reviewer #3 (Remarks to the Author):

Thank you for your responses.

One last comment: please add to the methods section a more detailed description of how exactly the adjustment for time/batch effect in microbiota analysis was done in fig.6.

A more detailed description of how the adjustment for time/batch has been added to the method section (Page; 24-25 Line; 501-506).

Thank you for the comment.